# THE BRAINY STUDENT: SCALABLE DEEP UNLEARNING BY SELECTIVELY DISOBEYING THE TEACHER

## ABSTRACT

Deep machine unlearning is the problem of removing the influence of a cohort of data from the weights of a trained deep model. This challenge has enjoyed increasing attention recently, motivated to the widespread use of neural networks in applications involving user data: allowing users to exercise their 'right to be forgotten' necessitates an effective unlearning algorithm. Deleting data from models is also of interest in practice for removing out-of-date examples, outliers or noisy labels. However, most previous unlearning methods consider simple scenarios where a theoretical treatment is possible. Consequently, not only do their guarantees not apply to deep neural networks, but they also scale poorly. In this paper, drawing inspiration from teacher-student methods, we propose a scalable deep unlearning method that breaks free of previous limiting assumptions. Our thorough empirical investigation reveals that our approach significantly improves upon previous methods in being by far the most *consistent* in achieving unlearning in a wide range of scenarios, while incurring only a minimal performance degradation, if any, and being significantly more scalable than previous methods.

## 1 INTRODUCTION

Can we make a deep learning model 'forget' a subset of its training data? Aside from being scientifically interesting, achieving this goal of Deep Machine Unlearning is increasingly important and relevant from a practical perspective. Regulations such as EU's General Data Protection Regulation (Mantelero, 2013) stipulate that individuals can request to have their data 'deleted' (the 'right to be forgotten'). Nowadays, given the ubiquity of deep learning systems in a wide range of applications, including computer vision, natural language processing, speech recognition and healthcare, allowing individuals to exercise this right necessitates deep unlearning algorithms. Further, in addition to privacy considerations, this type of data deletion may also be desirable for several practical applications: removing out-of-date examples, outliers, poisoned samples (Jagielski et al., 2018), noisy labels (Northcutt et al., 2021), or data that may carry harmful biases (Fabbrizzi et al., 2022).

However, truly removing the influence of a subset of the training set from the weights of a trained model is hard, since deep models memorize information about specific instances (Zhang et al., 2020; 2021; Arpit et al., 2017) and their highly non-convex nature makes it difficult to trace the effect of each example on the model's weights. Of course, we can apply the naive solution of re-training the model from scratch without the cohort of data to be forgotten. This conceptually-simple procedure indeed guarantees that the weights of the resulting model aren't influenced by the instances to forget (it performs 'exact unlearning'), but the obvious drawback is computational inefficiency: re-training a deep learning model to accommodate each new forgetting request isn't viable in practice.

To mitigate the large computational cost of exact unlearning, recent research has turned to *approximate unlearning* (Izzo et al., 2021; Golatkar et al., 2020a;b). The goal, as defined in these works, is to modify the weights of the trained model in order to produce a new set of weights that approximates those that would have been obtained by the exact procedure of re-training from scratch. That is, they strive to achieve 'indistinguishability' between the models produced by the exact and approximate solutions, typically accompanied by theoretical guarantees for the quality of that approximation.

Unfortunately, though, most previous approaches suffer from the following important problems. Firstly, their guarantees are derived in simple scenarios of convex loss functions, where theoretical analysis is feasible, thus excluding the arguably most relevant application of unlearning: deep neural

networks. Secondly, in light of recent evidence, model indistinguishability may not be a necessary nor sufficient condition for successful unlearning (Thudi et al., 2022; Goel et al., 2022), putting into question the soundness of these guarantees even in the cases where they do apply. Finally, previous approaches suffer from poor scalability, both to the size of the training dataset (and, consequently, the size of the models) as well as to the size of the cohort that will be forgotten (Goel et al., 2022).

The desire to scale to larger training datasets and model sizes needs little motivation. Indeed, this scaling has been a large contributing factor of the continuing progress of deep learning (Kaplan et al., 2020), creating an increasing need to also scale deep unlearning algorithms accordingly. In addition, scalability to the size of the forget set is very important since, as Goel et al. (2022) also point out, even a single user might own many samples from the training set that should be deleted, or several users may submit deletion requests in 'bursts' after certain events like revelations of privacy leakages by an organization. More generally, applications of removal of out-of-date or poisoned data, noisy labels or data that carries harmful biases don't come with assumptions on the size of those deletion sets. For instance, an organization may wish to delete all data that is older than a prescribed retention period. In all these cases, the forget sets can be very large.

Motivated by the above desiderata, we propose an approach that breaks free from limiting assumptions made in previous work and is scalable both to training set and forget set sizes. To that end, drawing inspiration from the teacher-student methodology, we propose an unlearning approach which we dub SCalable Remembering and Unlearning with a Brainy Student (SCRUBS). Concretely, we train a 'brainy' student model that is smart enough to disobey the teacher to avoid inheriting information about the forget cohort, while obeying it otherwise, to distill all useful information that it is allowed to keep. Our thorough empirical investigation reveals that our approach is by far the most *consistent* in achieving unlearning in a wide range of scenarios, while incurring only a minimal performance degradation, if any, and being significantly more scalable than previous methods.

## 2 PROBLEM DEFINITION

**Notation and preliminaries**  Let $\mathcal{D} = \{x_i, y_i\}_{i=1}^N$ denote a training dataset containing $N$ examples, where the $i$'th example is represented by a vector of input features $x_i$ and a corresponding class label $y_i$. Let $f(\cdot; w)$ denote a function parameterized by trainable weights $w$. In this work, we study the case where $f$ is represented by a deep neural network, trained in a supervised manner via empirical risk minimization. Specifically, we define the loss function as

$$\mathcal{L}(w) = \frac{1}{N} \sum_{i=1}^N l(f(x_i; w), y_i) \tag{1}$$

where $l$ denotes the cross entropy loss.

**Deep Machine Unlearning**  We now formalize the problem of deep machine unlearning. We assume that we are given a model $f(\cdot; w^o)$ that has been trained on $\mathcal{D}$, where $f$ denotes a neural network and $w^o$ its trained parameters, obtained by minimizing Equation 1. We will refer to this as the 'original model' (i.e. before any unlearning intervention is performed). We are also given a 'forget set' $\mathcal{D}_f = \{x_i, y_i\}_{i=1}^{N_f} \subset \mathcal{D}$, comprised of $N_f$ examples from $\mathcal{D}$ and a 'retain set' $\mathcal{D}_r$ of $N_r$ training examples that the model is still allowed to retain information for. For simplicity, we consider the standard scenario where $\mathcal{D}_r$ is the complement of $\mathcal{D}_f$ or, in other words, $\mathcal{D}_f \cup \mathcal{D}_r = \mathcal{D}$.

Given $f(\cdot; w^o)$, $\mathcal{D}_f$ and $\mathcal{D}_r$, the goal of deep machine unlearning is to design a scalable and efficient algorithm for producing a new set of weights $w^u$ that contains as little information as possible about $\mathcal{D}_f$ while not sacrificing performance. That is, $f(\cdot; w^u)$ and $f(\cdot; w^o)$ should perform equally well on the retain set, as well as on held-out test data outside of $\mathcal{D}$.

Formally, we want the following two equations to hold:

$$I(p(f(\mathcal{D}_f; w^u); p(f(\mathcal{D}_f; w^o))) = 0 \tag{2}$$
$$I(p(f(\mathcal{D}_r; w^u); p(f(\mathcal{D}_r; w^o))) = 1 \tag{3}$$

where $I(p; q)$ stands for the mutual information between the probability distributions of $p$ and $q$ and $p(f(\mathcal{D}; w))$ is the distribution over outputs produced by $f$ with parameters $w$ on the dataset $\mathcal{D}$.

Intuitively, Equation 2 states that the weights $w^u$ obtained from unlearning should yield a probability distribution that shares no information with that of the weights $w^o$ of the original model, when ingesting data from the forget set. On the other hand, Equation 3 states that those two probability distributions should agree on data from the retain set. Therefore, satisfying both Equations 2 and 3 simultaneously would lead to retaining all information about making predictions on examples from $\mathcal{D}_r$, while losing all information about making predictions on examples from $\mathcal{D}_f$.

**Trade-offs**  In practice, there may be trade-offs between three desiderata: scrubbing the weights clean of information from $\mathcal{D}_f$, maintaining accuracy on $\mathcal{D}_r$, and computational efficiency. For example, the naive solution of training from scratch on $\mathcal{D} - \mathcal{D}_f$ is a good solution with respect to the first two, but very poor with respect to the third. As another example, setting $w^u$ to be random weights indeed satisfies the first and third, but would suffer from very low performance on the second and, finally, not performing any unlearning violates the first but satisfies the second and third.

In agreement with Goel et al. (2022), we define our goal as searching for unlearning algorithms that live on the Pareto frontier of these three desiderata. However, different practitioners may have different preferences and priorities in weighting the amount of deletion versus the amount of performance degradation on the retain set, and we offer a knob to express these application-specific preferences in our framework through the hyperparameters in Equation 6 that we introduce in the next section.

## 3 SCALABLE REMEMBERING AND UNLEARNING WITH A BRAINY STUDENT (SCRUBS)

In this section, we present a practical algorithm for optimizing Equations 2 and 3, by casting the unlearning problem into a student-teacher framework. Specifically, we will think of the original model $w^o$ as the 'teacher' in this context, and we formulate our goal as training a 'student' model $w^u$ that *selectively* obeys that teacher. Intuitively, our goal for $w^u$ is twofold: unlearn $\mathcal{D}_f$ while still remembering $\mathcal{D}_r$. To that effect, $w^o$ should be obeyed when teaching about $\mathcal{D}_r$, but it should be disobeyed when teaching about $\mathcal{D}_f$. Consequently, we desire a 'brainy' student that is smart enough to selectively disobey the teacher when necessary.

En route to presenting our training objective, let us first define:

$$d(x; w^u) = D_{\mathrm{KL}}(p(f(x; w^o)) \| p(f(x; w^u)))$$

In words, $d(x; w^u)$ is the KL-divergence between the student and teacher distributions for the example $x$. We make the dependence of $d$ on $w^u$ explicit, since we will optimize the student weights $w^u$ while keeping the teacher weights frozen $w^o$, treating them as a constant.

Concretely, we begin by initializing the student $w^u$ to the weights of the teacher $w^o$. Since the teacher weights $w^o$ were trained on all of $\mathcal{D}$, we assume that the teacher performs well on both constituents of $\mathcal{D}$, $\mathcal{D}_r$ and $\mathcal{D}_f$. Therefore, this initialization step ensures that the student has good performance on $\mathcal{D}_r$ at initialization time, already fulfilling one of our desiderata. Can we then start from this solution and modify it to unlearn $\mathcal{D}_f$? In principle, one could do this by optimizing:

$$\min_{w^u} -\frac{1}{N_f} \sum_{x_f \in \mathcal{D}_f} d(x_f; w^u) \tag{4}$$

However, in practice, when performing this maximization of the distance between the student and teacher on the forget set, we noticed that, while indeed the performance on $\mathcal{D}_f$ degrades, as desired, it unfortunately also results in degrading performance on $\mathcal{D}_r$. To amend that, we propose to simultaneously encourage the student to 'stay close' to the teacher on retain examples, while encouraging it to 'move away' from the teacher on forget examples. Formally, the optimization objective becomes:

$$\min_{w^u} \frac{1}{N_r} \sum_{x_r \in \mathcal{D}_r} d(x_r; w^u) - \frac{1}{N_f} \sum_{x_f \in \mathcal{D}_f} d(x_f; w^u) \tag{5}$$

Finally, we empirically found it beneficial to also simultaneously optimize for the task loss on the retain set, to further strengthen the incentive to perform well there. Our final training objective is then the following:

$$\min_{w^u} \quad \frac{\alpha}{N_r} \sum_{x_r \in \mathcal{D}_r} d(x_r; w^u) + \frac{\gamma}{N_r} \sum_{(x_r, y_r) \in \mathcal{D}_r} l(f(x_r; w^u), y_r) - \frac{1}{N_f} \sum_{x_f \in \mathcal{D}_f} d(x_f; w^u) \quad (6)$$

where $l$ stands for the cross-entropy loss and the $\alpha$ and $\gamma$ are scalars that we treat as hyperparameters.

In practice, we found that optimizing the objective in Equation 6 is challenging, due to oscillations in the loss throughout training. Intuitively, this is due to the student aiming to simultaneously satisfy two objectives, which may interfere with each other. To address this, we provide a practical recipe for optimization, reminiscent of common 'tricks' used in other min-max problems like in Generative Adversarial Networks (GANs) (Goodfellow et al., 2014) where, in each iteration, the discriminator is trained for several steps before performing a single update to the generator. In our case, we iterate between performing an epoch of updates on the retain set (the *min-step*) followed by an epoch of updates on the forget set (the *max-step*), in an alternating fashion. We also found it helpful to perform a sequence of additional *min-steps* at the end of the sequence. For clarity, we provide pseudocode in Algorithm 1 in the Appendix and discuss these choices in Section 5.

## 4 RELATED WORK

**Unlearning definitions** Defining the problem of unlearning is nuanced and an open problem in and of itself. A popular definition of the goal of unlearning is to produce model weights which are indistinguishable from those of a model trained from scratch without the cohort to forget (Ginart et al., 2019; Graves et al., 2021; Guo et al., 2019; Golatkar et al., 2020a;b; 2021).

However, there is a recent line of criticism against this definition of unlearning. Thudi et al. (2022) theoretically show that we can obtain arbitrarily similar model weights from training on two non-overlapping datasets. This implies that the ability to achieve a particular set of parameters as the *end result* of training does not always speak to the training dataset that was used. To strengthen this conclusion, Shumailov et al. (2021) recently showed that it's possible to find different training examples that would yield a similar gradient for updating the model, due to the averaging operation over the minibatch when using stochastic gradient descent. Therefore, having an unlearning algorithm output parameters similar to those of a model trained from scratch isn't a sufficient condition for having achieved unlearning. Furthermore, Goel et al. (2022) argue that it's not a necessary condition either: by tweaking the training procedure, or changing hyperparameters, one will arrive at significantly different model distributions (Yang & Shami, 2020). Therefore, a successful unlearning method may be unnecessarily penalized for not matching a *particular* retrained-from-scratch model.

Inspired by this criticism, we instead opt for a different definition of the unlearning in our work, as presented in Section 2. Specifically, instead of using a model retrained from scratch as a reference point, we instead start from the *original model* trained on all of $\mathcal{D}$, and use that as a reference. Unlearning can then be defined in terms of removing information about $\mathcal{D}_f$ from that model while retaining the information it already contains about $\mathcal{D}_r$.

**Unlearning methods for convex models** The problem of machine unlearning was first introduced in (Cao & Yang, 2015), where they provide an exact forgetting algorithm for statistical query learning. (Bourtoule et al., 2021) proposed a training framework by sharding data and creating multiple models, enabling exact unlearning of certain data partitions. (Ginart et al., 2019) was the first paper to introduce a probabilistic definition for machine unlearning which was inspired by Differential Privacy (Dwork et al., 2014), and formed the origin of the idea of the model indistinguishability definition discussed above. More recently, Guo et al. (2019); Izzo et al. (2021); Sekhari et al. (2021) built upon this framework and introduced unlearning methods that can provide theoretical guarantees under certain assumptions.

**Unlearning methods for deep networks** Recently, approximate unlearning methods were developed that can be applied to deep neural networks. (Golatkar et al., 2020a) proposed an information-theoretic procedure for removing information about $\mathcal{D}_f$ from the weights of a neural network and (Golatkar et al., 2020b; 2021) propose methods to approximate the weights that would have been obtained by unlearning via a linearization inspired by NTK theory (Jacot et al., 2018) in the first case, and based on Taylor expansions in the latter.

| Model | Forget 3 classes (50%) | | | | | | Forget 4 classes ($\sim$ 67%) | | | | | |
|---|---|---|---|---|---|---|---|---|---|---|---|---|
| | Test error ($\downarrow$) | | Retain error ($\downarrow$) | | Forget error ($\uparrow$) | | Test error ($\downarrow$) | | Retain error ($\downarrow$) | | Forget error ($\uparrow$) | |
| | mean | std | mean | std | mean | std | mean | std | mean | std | mean | std |
| retrain | **32.2** | 2.22 | 0.0 | 0.0 | **99.9** | 0.19 | **14.5** | 2.18 | 0.0 | 0.0 | **99.8** | 0.14 |
| Original | 37.7 | 3.0 | 0.0 | 0.0 | 0.0 | 0.0 | 33.7 | 1.04 | 0.0 | 0.0 | 0.0 | 0.0 |
| Finetune | **33.2** | 3.0 | 0.0 | 0.0 | 4.9 | 0.5 | 20.3 | 2.75 | 0.0 | 0.0 | 9.2 | 0.9 |
| Fisher | 37.4 | 2.7 | 15.4 | 7.6 | 84.4 | 6.4 | 21.8 | 7.09 | 7.83 | 9.09 | 84.3 | 2.8 |
| NTK | 35.2 | 2.9 | 0.0 | 0.0 | 25.7 | 22.3 | 18.5 | 3.8 | 0.0 | 0.0 | 46.9 | 5.9 |
| NegGrad | **31.1** | 2.2 | 1.2 | 1.5 | 89.2 | 1.8 | **12.8** | 1.8 | 0.0 | 0.0 | 78.8 | 1.4 |
| CF-k | **32.6** | 3.1 | 0.0 | 0.0 | 8.2 | 1.0 | 18.8 | 2.4 | 0.0 | 0.0 | 14.3 | 1.2 |
| EU-k | **32.6** | 3.9 | 0.2 | 0.2 | 55.4 | 2.2 | **15.7** | 1.8 | 0.2 | 0.3 | 64.3 | 0.7 |
| SCRUBS | **30.8** | 1.3 | 0.0 | 0.0 | **98.6** | 2.0 | **15.3** | 2.3 | 0.0 | 0.0 | **99.6** | 0.7 |

Table 1: For **large forget sets, class unlearning** on CIFAR-6 (with ResNet), SCRUBS achieves significantly better forgetting than all alternatives and doesn't degrade performance.

However, (Golatkar et al., 2020a;b) scale poorly with the size of the training dataset, as computing the forgetting step scales quadratically with the number of training samples. (Golatkar et al., 2021) addresses this issue, albeit under a different restrictive assumption. Specifically, they assume that a large dataset is available for pre-training that will remain 'static', in the sense that no forgetting operations are expected to be applied on it. Unfortunately, this assumption is not met in many important realistic applications of unlearning, like in healthcare where, for instance, we can't assume availability of a large static dataset of medical images specific to each problem of interest. (Chundawat et al., 2022) is a recent preprint that explores another teacher-student framework for unlearning. We include a section in the Appendix where we extensively discuss the differences with it and empirically compare against it.

**Related research areas**  Differential privacy (Dwork et al., 2014) and life-long learning (Parisi et al., 2019) have shared goals with unlearning. Differential Privacy seeks to ensure that the trained model doesn't store information about *any* training instance; a strong goal that is difficult to achieve for deep neural networks (Abadi et al., 2016). In contrast, in machine unlearning, we only desire to remove the influence of the training instances in the given forget set, which can be seen as a relaxation of the same problem. Life-long learning seeks to continuously update a model to solve new sets of tasks, without 'catastrophically forgetting' previously-learned tasks. The notion of forgetting a task, though, is distinct from that of forgetting a training example: the performance on a task can degrade, without having removed influence of particular training examples from the model weights.

**Related training objectives**  The teacher-student methodology has been used in a variety of contexts, including self-supervised (Grill et al., 2020; Chen & He, 2021) and semi-supervised learning (Xie et al., 2020). A related idea is Knowledge Distillation (Hinton et al., 2015), which can be thought of as a teacher-student approach with the additional desire for compression, where the student is chosen to have a smaller architecture than the teacher. In addition, our training objective has a contrastive flavour (Chen et al., 2020; He et al., 2020), due to both pulling the student close to the teacher, and pushing it away from it, for different parts of the training set.

## 5   EXPERIMENTS

The goal of our experimental investigation is to answer the following questions: **Q1**: How do different methods perform for the important understudied case of larger forget sets? **Q2**: How does SCRUBS fare against the state-of-the-art models in the established benchmarks? **Q3**: How do different methods fare in terms of *consistency* of performance across different scenarios (datasets, architectures, training and forget set sizes, class vs selective forgetting)? **Q4**: How fast do the different methods run, compared to the assumed-to-be prohibitively-expensive retraining from scratch?

### 5.1   EXPERIMENTAL SETUP

**Methods**  We compare against state-of-the-art approaches as well as various baselines:

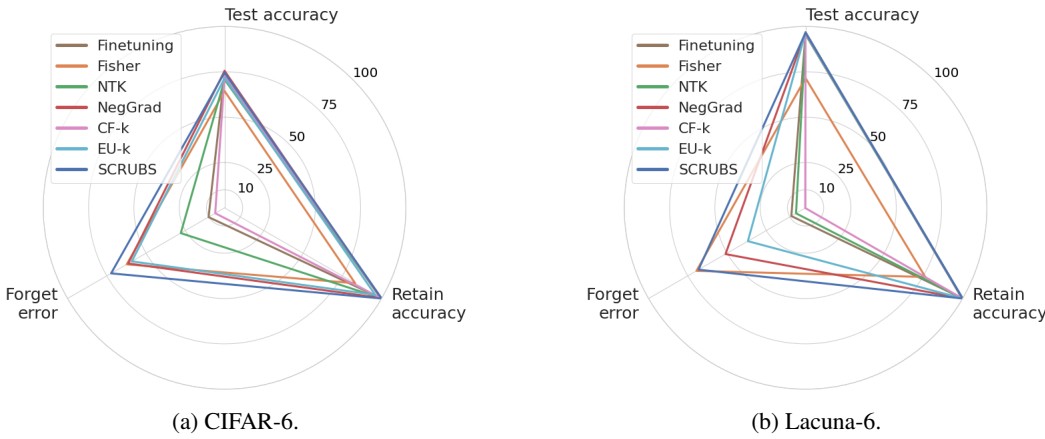

(a) CIFAR-6.                                    (b) Lacuna-6.

Figure 1: In **large forget sets**, SCRUBS is the only top-performer across settings. Each point represents the average across 6 settings: {ResNet, All-CNN} x {class-50%, class-67%, selective-50%}, using the data from Tables 1, 11, 12, 13, 14 and 15.

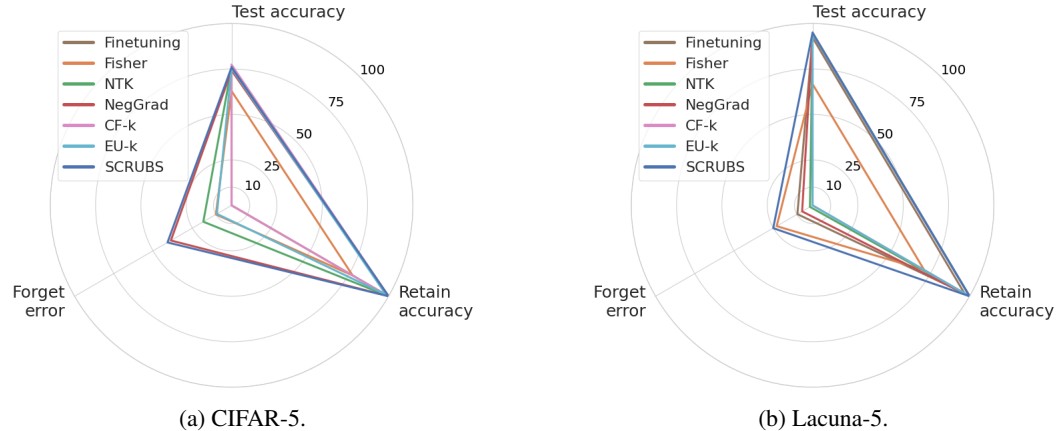

(a) CIFAR-5.                                    (b) Lacuna-5.

Figure 2: In the **small scale setting, selective unlearning**, SCRUBS is the only top-performer across settings. Each point represents the average across ResNet and All-CNN. On the left, CF-k and Finetuning perform similarly (they perform poorly) so their lines occlude one another, and the same is true of NTK, CF-k and EU-k on the right. For reference, this data lives in Tables 4 and 5.

- **Retrain**: Retraining from scratch without the cohort to forget. This method is assumed to not be viable in practice due to its prohibitive computational cost.

- **Original**: The 'original' model trained on all of $\mathcal{D}$ without any unlearning intervention.

- **Finetuning**: Finetunes the original model on data from the retain set $\mathcal{D}_r$.

- **NegGrad**: Similar to Finetuning, starts from the original model and finetunes it both on data from the retain set and the forget set, negating the gradient for the latter. Previous work considered a weaker baseline that only trains on the forget set, with a negative gradient. We tune this stronger baseline to achieve a good balance between the two objectives.

- **Fisher**: The Fisher Forgetting method from (Golatkar et al., 2020a).

- **NTK**: The NTK Forgetting method from (Golatkar et al., 2020b).

- **CF-k**: Catastrophic Forgetting-k (CF-k) (Goel et al., 2022) freezes the first (bottom-most) k layers of the original model and finetunes the remaining (top-most) layers on $\mathcal{D}_r$.

- **EU-k**: Exact Unlearning-k (EU-k) (Goel et al., 2022) freezes the first (bottom-most) k layers of the original model and trains from scratch the remaining (top-most) layers on $\mathcal{D}_r$.

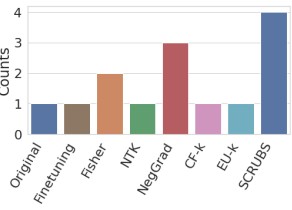 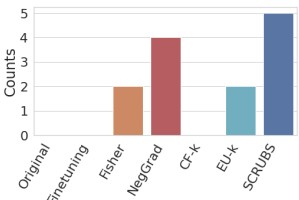 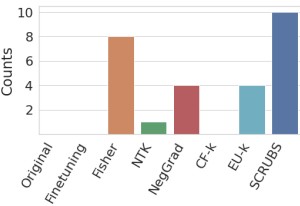

(a) Small scale (4 settings).  (b) Large scale (8 settings).  (c) Large forget sets (12 settings).

Figure 3: The number of times that each method was (one of the) top performer(s) in terms of the forget error, across all settings (architectures, datasets, selective vs class forgetting, forget set size, etc.). We consider a model a top-performer if its 95% confidence interval overlaps with that of the best mean. Small scale counts are over {ResNet, All-CNN} x {CIFAR, Lacuna} (4 total), large scale additionally x {class, selective} (8 total), and large forget set additionally x {class-50%, class-67%, selective-50%} (12 total). **SCRUBS is the most consistent in terms of its ability to unlearn**.

**Metrics**   We desire the model to have forgotten about $\mathcal{D}_f$, without hurting its performance on the retain set nor its generalization ability. We therefore report **forget error**, where higher values are better (denoted $\uparrow$ in the tables), and **retain error** and **test error** where lower values are better (denoted $\downarrow$ in the tables). For larger-scale experiments, we also consider **scale-up factor** as an additional metric: the fraction of the runtime of retraining from scratch over the runtime of the given unlearning algorithm. Finally, we posit that *consistency* in good performance on the above metrics is crucial for practical applicability, so we experiment with a large number of settings, including different datasets, sizes of forget and training sets, architectures and selective or class forgetting.

**Experimental setup**   We base our investigation on the same two datasets that are used in previous work: CIFAR-10 (Krizhevsky et al., 2009) and Lacuna-10 (Golatkar et al., 2020a), which is a dataset derived from VGG-Faces (Cao & Yang, 2015). We also use the same two architectures as in previous work: All-CNN (Springenberg et al., 2014) and ResNet-18 (He et al., 2016). For fair comparisons with previous work, we follow the standard setup of using a model that was pre-trained on CIFAR-100 / Lacuna-100 for the CIFAR-10 / Lacuna-10 experiments, respectively (Golatkar et al., 2020a;b). We run each experiment with 3 random seeds and report the mean and standard deviation. We utilize the public code for the NTK and Fisher methods in our experiments, to ensure correctness. We also plan to release our own code upon publication to facilitate future comparisons.

We conduct our empirical investigation in 3 setups, described in further detail in the Appendix:

- **Large forget set sizes**: We significantly increase the % of the training set that belongs to $\mathcal{D}_f$; an axis orthogonal to that of increasing the training set size. We investigate forgetting up to $\sim 67\%$ of the training set (for reference, standard benchmarks consider up to 10%). For this, we train on 6 classes from each of CIFAR and Lacuna ('CIFAR-6' / 'Lacuna-6') and perform both *selective unlearning*, where we forget 50% of the training examples of each class, and *class unlearning* where we forget either 3 (50%) or 4 classes ($\sim 67\%$).

- **Small-scale**: We exactly follow the setup in (Golatkar et al., 2020b) that uses only 5 classes from each of CIFAR and Lacuna ('CIFAR-5' / 'Lacuna-5'), with 100 train, 25 validation and 100 test examples per class. The forget set contains 25 examples from class 0 (5%). This small-scale setting allows comparing to NTK, which doesn't scale to larger datasets.

- **Large-scale**: We exactly follow the setup in (Golatkar et al., 2020a) that uses all 10 classes of each of CIFAR and Lacuna, and considers both a *class unlearning* scenario where the forget set is the entire training set for class 5 (10%), as well as a *selective unlearning* one where 100 examples of class 5 are forgotten (0.25% in CIFAR and 2% in Lacuna).

## 5.2   FINDINGS AND TAKE-AWAYS

We present our results on exploring larger forget set sizes (**Q1**) in Table 1 (and Tables 11, 12, 13, 14, 15), and Figure 1 (and Figures 7, 8). In investigating the competitiveness of SCRUBS on previously-established benchmarks (**Q2**), we report results for the small-scale in Figure 2 (and Tables 4 and 5)

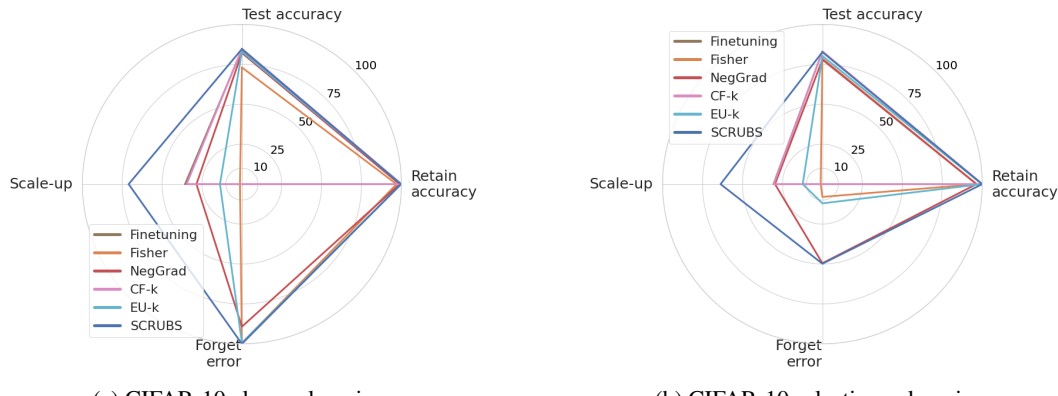

(a) CIFAR-10 class unlearning.        (b) CIFAR-10 selective unlearning.

Figure 4: In the **large-scale setting**, SCRUBS is the only top-performer across settings. Selective forgetting is much harder for all, but especially for EU-k and Fisher. Scale-up is multiplied by 10 for visualization purposes. Each point represents the average across ResNet and All-CNN variants. On both subplots, CF-k and Finetuning perform similarly (they perform poorly) so their lines occlude one another. For reference, the data lives in Tables 6 and 7 (left) and Tables 8 and 9 (right).

and for the large-scale in Figure 4 (and Tables 6, 7, 8, 9). We showcase the consistency of each method's ability to unlearn (**Q3**) in Figure 3, and we report results for the scale-up factor in Figures 4 (and Figure 6) (**Q4**). Tables and Figures mentioned in brackets can be found in the Appendix.

We summarize our main findings below.

- **SCRUBS** is the sole method that is *consistently* a top-performer in terms of achieving unlearning (Figure 3), while incurring a minimal performance degradation, if any (Table 1, Figures 1, 2 and 4), and being very computationally efficient (Figure 4).

- **Original**, as expected, enjoys good performance on the retain and test sets, but it has low forget set error, reflecting that no effort was made to unlearn $\mathcal{D}_f$.

- **Finetuning** retains the performance of Original for the most part, but also fails at forgetting.

- The previous state-the-art **NTK** isn't among the top-performers in terms of forgetting, though it at least doesn't hurt the model's performance (Figures 1 and 2). However, our thorough investigation of larger forget set sizes revealed a notable exception to its poor performance, on selective unlearning with large forget set sizes (Figure 8). This warrants further investigation to gain a deeper understanding of NTK, though it isn't viable in practice since it can't scale beyond very small datasets and performs poorly on several settings.

- The previous state-the-art **Fisher** sometimes achieves forgetting (Figures 1b, 14b and 4a, 8), though not consistently (Figure 3), and unfortunately usually degrades the model's performance substantially. It is also very slow (Figure 4), actually exceeding the runtime of the retrain from scratch; an observation also made in (Goel et al., 2022).

- The baseline of **NegGrad** turns out to be a strong one in terms of achieving a balance between unlearning and retaining performance and we encourage future work to also report this baseline. However, SCRUBS outperforms it in terms of forget error in several scenarios, as seen in Figure 3 and is much more computationally efficient (see Figure 4).

- **CF-k** completely fails to forget in most cases, whereas **EU-k** is more successful in that regard (though not consistently (Figure 3)). This is expected, since EU-k trains the top layers from scratch, thus increasing the chances of removing information about $\mathcal{D}_f$ whereas CF-k finetunes those layers instead. A notable failure case that we discovered for EU-k is *selective* forgetting (notice the contrast between EU-k's forget error between Figures 4a and 4b and between Figures 7b and 8b). This interesting finding may speak to where class vs instance information is stored in neural networks and warrants further investigation.

**Analysis of Training Dynamics**  Next, we illustrate the training dynamics of SCRUBS and the importance of different design choices. As a reminder, the student is initialized from the teacher and

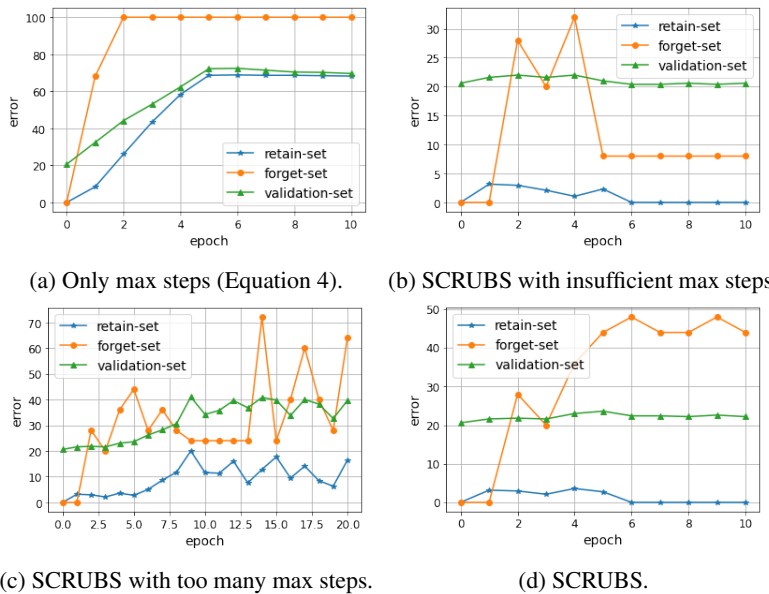

(a) Only max steps (Equation 4).  (b) SCRUBS with insufficient max steps.

(c) SCRUBS with too many max steps.  (d) SCRUBS.

Figure 5: Illustration of training dynamics of SCRUBS variants, on CIFAR-5 with a ResNet model. Performing the right interleaving of *min-steps* and *max-steps* is important for achieving a good balance between high forget error and low retain and validation errors.

subsequently undergoes an alternating sequence of *max-steps* and *min-steps*; the former encouraging the student to move far from the teacher on the forget set, and the latter encouraging it to stay close to the teacher on the retain set. We also found it useful to perform a sequence of additional *min-steps* after the alternating sequence. We now explore the effect of these decisions.

First, we show that performing only *max-steps*, by optimizing Equation 4, is not a good solution. Simply pushing the student away from the teacher on the forget set achieves forgetting but unfortunately also hurts the retain and validation set performance (Figure 5a). Therefore, alternating between *max-steps* and *min-steps* is necessary. However, it is important to find the right balance. For instance, as seen in Figure 5b, performing too few *max-steps* leads to the unwanted consequence of the forget error dropping. On the other hand, removing the final sequence of only *min-steps* is also harmful, as shown in Figure 5c that trains for a larger number of epochs of an equal number of (alternating) *max-steps* and *min-steps* without achieving a good balance at any point throughout the trajectory. On the other hand, SCRUBS (Figure 5d) achieves a good balance of high forget error and low retain and validation error simultaneously. The Appendix shows additional examples of training dynamics (Figures 9, 10, 11), hyperparameter sensitivity (Table 21), possible trade-offs of forgetting-versus-performance (Table 20) and an ablation of the cross-entropy term in Equation 6, which provides a small but consistent added protection against degrading performance (Figure 12).

## 6 CONCLUSION

In this paper, we have proposed SCRUBS, a new approach to deep machine unlearning, wherein a 'brainy' student selectively obeys the teacher model, in order to distill useful information from it, while at the same time erasing information pertaining to the cohort to forget. Our thorough empirical investigation revealed that SCRUBS is by far the most consistent approach in achieving forgetting, while only incurring a minimal performance degradation, if any, in all scenarios we considered, and being significantly more computationally efficient than previous methods. Areas for future work include developing practical algorithms to improve the stability of optimization of SCRUBS and proving theoretical guarantees for the mutual information bounds. Overall, compared to previous works, we take a practical perspective and propose a strong-performing model that is also very computationally-efficient and a consistent top-performer across a wide range of scenarios. We hope that our work opens the door to practical applications of unlearning, enabling efficient deletions of (any amount of) out-of-date, noisy, or outlier data in large models, trained on large datasets.

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

# A APPENDIX

## A.1 DISCUSSION OF LIMITATIONS AND ADDITIONAL FUTURE WORK

SCRUBS has shown impressive results in terms of being consistently a top-performer in terms of unlearning, with a minimal drop in performance compared to previous works. However, SCRUBS has limitations that we hope future work will address. Primarily, the lack of theoretical guarantees is a limitation of our method. While this is an important drawback, theoretical guarantees are challenging for deep neural networks. Previous works associated with guarantees, despite offering great insights, suffer from practical limitations. We aim to fill this gap instead. However, we look forward to future work that strives to strike a compromise between effective unlearning, good performance, scalability, and theoretical insights. Another limitation of SCRUBS is the difficulty and instability associated with tuning the min-max objective, as shown in the literature e.g. for GANs. For instance, this can lead to oscillating behaviour, as we show in Figure 5. We remedy this to a large extent in practice by providing a practical algorithm that works well, showing consistently improved results over prior work, but there is room for improvement on this front for future work.

We hope that future work also continues to push the limits of scalability. In this work, we have made important strides in this direction. However, the datasets and models we consider aren't too large, in order to allow comparisons to previous works that would not be feasible to run for larger scale experiments. An interesting topic of future work is investigating the interplay between SCRUBS and other scalable algorithms like NegGrad with increasing amounts of scale.

Further, our extensive investigation has revealed interesting trade-offs between different models. However, a solid understanding of the stengths and weaknesses of each, as well as which factors impact their performance the most, remains elusive. For instance, our investigation revealed a scenario (selective unlearning in cases with large forget set sizes) where NTK and Fisher, albeit performing very poorly in most cases, perform uncharacteristically well (see e.g. Figure 8). This is a surprising finding that we've uncovered, in the sense that previous knowledge could not have trivially predicted it. Therefore, we hope that future work carefully studies existing methods to gain a deeper understanding of them.

Another really interesting future work direction is to investigate how different unlearning algorithms interact with different architectures, like Transformers, and loss functions, like self-supervised learning. In addition, future work can explore the effect of class imbalance on unlearning. For instance, is it easier to unlearn a class that has few training examples, compared to a class that has many training examples? Do the evaluation metrics need to be adapted for this scenario?

## A.2 MORE EXPERIMENTAL DETAILS

**Datasets**. We have used CIFAR-10 and Lacuna-10 datasets for evaluation purposes. CIFAR-10 consists of 10 classes with 60000 color images of size 32 x 32. In our experiments, the train, test, and validation sizes are 40000, 10000, and 10000 respectively. Lacuna-10 is a dataset derived from VGG-Faces (Cao & Yang, 2015). We have followed the same procedure described in (Golatkar et al., 2020a) to build Lacuna. We randomly select 10 celebrities (classes) with at least 500 samples. We use 100 samples of each class to form the test-set, and the rest make the train-set. All the images are resized to 32 x 32. We also use CIFAR-100 and Lacuna-100 to pretrain the models. Lacuna-100 is built in a similar way as Lacuna-10, and there is no overlap between the two datasets. We have not applied any data augmentation throughout the experiments.

**Small-Scale datasets**. We followed the same procedure as decribed in (Golatkar et al., 2020b) to create the small versions of CIFAR-10 and Lacuna-10, namely CIFAR-5 and Lacuna-5. To this end, we take the first 5 classes of each dataset and randomly sample 100 images for each class. We make the train and test sets by sampling from the respective train and test sets of CIFAR-10 and Lacuna-10. We also make 25 samples from each class from the train set to create the validation sets. For large forget-set experiments, CIFAR-6 and Lacuna-6, we repeat the same process by taking the first 6 classes of each dataset.

**Metrics** Here, we define mathematically the metrics used in our tables: retain error, forget error and test error. Let $\mathcal{D}_r$, $\mathcal{D}_f$ and $\mathcal{D}_t$ denote the retain and forget portions of the training dataset, and a test dataset of heldout examples, respectively. We define error ($Err$) as follows:

$$Err(\mathcal{D}) = 1 - \frac{1}{|\mathcal{D}|} \sum_{(x_i, y_i) \in \mathcal{D}} \mathbb{1}[\arg\max(f(x_i; w)) == y_i] \tag{7}$$

where $f$, parameterized by $w$ is the neural network model (comprised of a feature extractor followed by a softmax classifier layer), $\arg\max(f(x_i; w))$ is the label that the model thinks is most likely for example $x_i$, and $\mathbb{1}[x]$ is the indicator function that returns 1 if $x$ is True and 0 otherwise.

Based on the above, the retain error, forget error and test error are computed as $Err(\mathcal{D}_r)$, $Err(\mathcal{D}_f)$ and $Err(\mathcal{D}_t)$, respectively.

**Models.** We use the same models with the same architectural modifications in (Golatkar et al., 2020a;b). For All-CNN, the number of layers is reduced and batch normalization is added before each non-linearity. For Resnet, ResNet-18 architecture is used. For small scale experiments, the number of filters is reduced by 60% in each block. For the large-scale experiments, the exact architecture has been used.

**Pretraining.** Following the previous work for consistency, we apply pretraining. Specifically, for CIFAR datasets, we have pretrained the models on CIFAR-100. For Lacuna, we have pretrained the models on Lacuna-100. We pretrain the models for 30 epochs using SGD with a fixed learning rate of 0.1, Cross-Entropy loss function, weight decay 0.0005, momentum 0.9, and batch size 128.

**Baselines.** 'Original' is the model trained on the entire dataset $D$. For 'Retrain', we train the same architecture on $D_r$, with the same hyperparameters used during training of the original model. For 'Finetune', we fine-tune the 'original' model on $D_r$ for 10 epochs, with a fixed learning rate of 0.01 and weight-decay 0.0005. For 'Negative Gradient', we fine-tune the 'original' model using the following loss:

$$\mathcal{L}(w) = \beta \times \frac{1}{|D_r|} \sum_{i=1}^{|D_r|} l(f(x_i; w), y_i) - (1 - \beta) \times \frac{1}{|D_f|} \sum_{j=1}^{|D_f|} l(f(x_j; w), y_j) \tag{8}$$

Where $\beta \in [0, 1]$. We have tuned $\beta$ to get a high forget-error while not destroying retain-error. For small-scale experiments, $\beta = 0.95$ and we have trained for 10 epochs, with SGD, 0.01 lr and 0.1 weight-decay. For large-scale experiments $\beta = 0.9999$ and we have trained for 5 epochs, with SGD, 0.01 lr, and 0.0005 weight-deay. Please note that small $\beta$ result in explosion quickly. For 'CF-k', we freeze the first k layers of the network and finetune the rest layers with $D_r$. We use the same setting as 'Finetune' baseline. For 'EU-k' we freeze the first k layers, and re-initialize the weights of the remaining layers and retrain them with $D_r$. As all the models are pretrained on larger datasets, for re-initializing we use the weights of the pretrained models. For 'EU-k' we use the same settings as the 'Retrain' baseline. In both 'EU-k' and 'CF-k' baselines, for both ResNet and All-CNN we freeze all the layers except for the last block of the network. For Resnet the last block is block4 and for All-CNN, the last block of layers is the 9th sequential block.

**SCRUBS parameters.** We train SCRUBS using Algorithm 1. Throughout the experiments, we tune the parameters to get a high forget-error while retaining the retain and validation error of the original model. We use the same optimizer for both min and max steps. We observed that for small-scale settings 'Adam' optimizer works better, while in large-scale settings both 'Adam' and 'SGD' could be used. For all experiments, we initialize the learning rate at 0.0005 and decay it by 0.1 after a number of min and max steps. Decaying the learning rate is crucial to control the oscillating behaviour of our min and max optimization. We apply a weight decay of 0.1 for small-scale setting and 0.0005 for large scale experiments, with a momentum of 0.9. Finally, we use different batch sizes for the forget-set and the retain-set to control the number of iteration in each direction, i.e the max and the min respectively. We report these in Table 2.

**System specification.** For scale-up experiments, All the codes are written and executed in Python 3.8, on an Ubuntu 20 machine with 40 CPU cores, a Nvidia GTX 2080 GPU and 256GB memory.

## A.3 PSEUDOCODE

For clarity, we provide pseudocode for training SCRUBS in Algorithm 1. We also plan to make our code public upon acceptance to facilitate future research.

| model | dataset | unlearning-type | forget-set bs | retain-set bs | max steps | min steps |
|-------|---------|-----------------|---------------|---------------|-----------|-----------|
| ResNet | CIFAR-10 | class | 512 | 128 | 2 | 3 |
| | CIFAR-10 | selective | 16 | 64 | 5 | 5 |
| | Lacuna-10 | class | 128 | 128 | 5 | 5 |
| | Lacuna-10 | selective | 32 | 32 | 4 | 4 |
| | CIFAR-5 | selective | 32 | 32 | 10 | 10 |
| | Lacuna-5 | selective | 32 | 32 | 5 | 10 |
| | CIFAR-6 | class | 64 | 32 | 2 | 10 |
| | CIFAR-6 | selective | 64 | 32 | 5 | 10 |
| | Lacuna-6 | class | 32 | 32 | 2 | 10 |
| | Lacuna-6 | selective | 64 | 32 | 4 | 8 |
| All-CNN | CIFAR-10 | class | 512 | 256 | 3 | 4 |
| | CIFAR-10 | selective | 16 | 64 | 5 | 5 |
| | Lacuna-10 | class | 32 | 32 | 4 | 4 |
| | Lacuna-10 | selective | 8 | 32 | 2 | 4 |
| | CIFAR-5 | selective | 16 | 32 | 5 | 10 |
| | Lacuna-5 | selective | 32 | 32 | 5 | 10 |
| | CIFAR-6 | class | 32 | 32 | 2 | 10 |
| | CIFAR-6 | selective | 64 | 32 | 3 | 10 |
| | Lacuna-6 | class | 32 | 32 | 2 | 10 |
| | Lacuna-6 | selective | 32 | 32 | 3 | 10 |

Table 2: SCRUBS hyperparameters for each experiment

---

**Algorithm 1** Training a Brainy Student

---

**Require:** Teacher weights $w^o$
**Require:** Total max steps MAX-STEPS
**Require:** Total steps STEPS
**Require:** Learning rate $\epsilon$
  $w^u \leftarrow w^o$
  $i \leftarrow 0$
  **while** $i <$ STEPS **do**
    **if** $i <$ MAX-STEPS **then**
      $w^u \leftarrow$ DO-MAX-EPOCH$(w^u)$
    **end if**
    $w^u \leftarrow$ DO-MIN-EPOCH$(w^u)$
  **end while**

---

**Algorithm 2** DO-MAX-EPOCH

---

**Require:** Student weights $w^u$
**Require:** Learning rate $\epsilon$
**Require:** Batch size B
**Require:** Forget set $D_f$
**Require:** Procedure NEXT-BATCH
  $b \leftarrow$ NEXT-BATCH$(D_f, \text{B})$
  **while** $b$ **do**
    $w^u \leftarrow w^u + \epsilon \nabla_{w^u} \dfrac{1}{|b|} \sum_{x_f \in b} d(x_f; w^u)$
    $b \leftarrow$ NEXT-BATCH$(D_f, \text{B})$
  **end while**

---

---

**Algorithm 3** DO-MIN-EPOCH

---

**Require:** Student weights $w^u$
**Require:** Learning rate $\epsilon$
**Require:** Batch size B
**Require:** Retain set $D_r$
**Require:** Procedure NEXT-BATCH
  $b \leftarrow \text{NEXT-BATCH}(D_r, \text{B})$
  **while** $b$ **do**
$$w^u \leftarrow w^u - \epsilon \nabla_{w^u} \frac{1}{|b|} \sum_{(x_r, y_r) \in b} \alpha d(x_r; w^u) + \gamma l(f(x_r; w^u), y_r)$$
    $b \leftarrow \text{NEXT-BATCH}(D_r, \text{B})$
  **end while**

---

|  | CIFAR-10 | | | | Lacuna-10 | | | |
|---|---|---|---|---|---|---|---|---|
|  | ResNet | | All-CNN | | ResNet | | All-CNN | |
| Model | all | selective | all | selective | all | selective | all | selective |
| Finetune | 3.8 | 3.09 | 3.33 | 3.03 | 1.7 | 2.03 | 2.16 | 2.00 |
| Fisher | 0.08 | 0.07 | 0.16 | 0.14 | 0.08 | 0.07 | 0.16 | 0.15 |
| NegGrad | 3.4 | 2.96 | 2.30 | 2.97 | 1.66 | 1.5 | 2.41 | 2.27 |
| CF-k | 3.55 | 3.17 | 3.37 | 2.91 | 3.42 | 3.20 | 3.27 | 3.11 |
| EU-k | 1.41 | 1.26 | 1.34 | 1.20 | 1.39 | 1.28 | 1.32 | 1.26 |
| SCRUBS | 7.84 | 7.41 | 6.36 | 5.33 | 2.17 | 1.95 | 2.81 | 2.48 |

Table 3: **Scale-up factor**: the fraction of the runtime of retrain from scratch over the runtime of each given unlearning algorithm. That is, a scale-up value of X for an unlearning algorithm means that that algorithm runs X times faster than retrain from scratch.

## A.4 ADDITIONAL TABLES

In this section, we provide the results for all scenarios we studied, for completeness. Each of these tables has been used in aggregated results reported in the main paper (e.g. in the barplot of counts of each method being a top-performer w.r.t forget error, in Figure 3, as well as in the spider plots where averages across settings are reported, in Figures 7, 2, 4).

Overall, our thorough investigation reveals interesting trade-offs between methods, summarized in Section 5 in the main paper. Our broad take-away is that SCRUBS is the most consistent in achieving unlearning, while causing only a minimal, if any, performance degradation and being very computationally-efficient.

|  | CIFAR-5 | | | | | | Lacuna-5 | | | | | |
|---|---|---|---|---|---|---|---|---|---|---|---|---|
|  | Test error ($\downarrow$) | | Retain error ($\downarrow$) | | Forget error ($\uparrow$) | | Test error ($\downarrow$) | | Retain error ($\downarrow$) | | Forget error ($\uparrow$) | |
| Model | mean | std | mean | std | mean | std | mean | std | mean | std | mean | std |
| Retrain | 24.9 | 2.5 | 0.0 | 0.0 | 28.8 | 5.9 | 5.8 | 0.4 | 0.0 | 0.0 | 4.8 | 3.4 |
| Original | 24.2 | 2.6 | 0.0 | 0.0 | 0.0 | 0.0 | 5.7 | 0.4 | 0.0 | 0.0 | 0.0 | 0.0 |
| Finetune | 24.3 | 2.4 | 0.0 | 0.0 | 0.0 | 0.0 | 5.6 | 0.3 | 0.0 | 0.0 | 0.0 | 0.0 |
| Fisher | 31.6 | 3.4 | 14.0 | 6.0 | 4.8 | 5.2 | 14.0 | 3.6 | 6.7 | 3.3 | 6.4 | 8.3 |
| NTK | 24.4 | 2.6 | 0.0 | 0.0 | 22.4 | 9.2 | 5.6 | 0.5 | 0.0 | 0.0 | 0.0 | 0.0 |
| NegGrad | 25.5 | 1.1 | 0.0 | 0.0 | 41.3 | 6.1 | 6.1 | 0.7 | 0.0 | 0.0 | 1.3 | 2.3 |
| CF-k | 22.6 | 1.9 | 0.0 | 0.0 | 0.0 | 0.0 | 5.8 | 0.4 | 0.0 | 0.0 | 0.0 | 0.0 |
| EU-k | 23.5 | 1.1 | 0.0 | 0.0 | 10.7 | 2.3 | 5.9 | 0.6 | 0.0 | 0.0 | 0.0 | 0.0 |
| Bad-T | 29.93 | 2.34 | 7.44 | 1.79 | 9.33 | 6.11 | 13.6 | 0.57 | 1.58 | 1.34 | 0.0 | 0.0 |
| SCRUBS | 24.2 | 1.6 | 0.0 | 0.0 | 40.8 | 1.8 | 6.2 | 0.73 | 0.0 | 0.0 | 24.8 | 5.2 |

Table 4: **Small-scale** results with ResNet. SCRUBS is the top-performer in terms of forgetting with minimal performance degradation.

| Model | CIFAR-5 | | | | | | Lacuna-5 | | | | | |
|---|---|---|---|---|---|---|---|---|---|---|---|---|
| | Test error (↓) | | Retain error (↓) | | Forget error (↑) | | Test error (↓) | | Retain error (↓) | | Forget error (↑) | |
| | mean | std | mean | std | mean | std | mean | std | mean | std | mean | std |
| Retrain | 24.36 | 1.61 | 0.13 | 0.28 | 28.8 | 9.12 | 4.6 | 0.38 | 0.0 | 0.0 | 4.67 | 6.41 |
| Original | 24.08 | 1.86 | 0.17 | 0.38 | 0.0 | 0.0 | 4.53 | 0.47 | 0.0 | 0.0 | 0.0 | 0.0 |
| Finetune | 23.48 | 1.91 | 0.04 | 0.09 | 0.0 | 0.0 | 9.77 | 10.76 | 6.63 | 13.22 | 19.33 | 40.03 |
| Fisher | 42.64 | 6.56 | 31.83 | 10.47 | 15.2 | 16.83 | 52.53 | 13.87 | 51.09 | 14.54 | 39.33 | 40.43 |
| NTK | 24.16 | 1.77 | 0.17 | 0.38 | 13.6 | 8.29 | 4.47 | 0.47 | 0.0 | 0.0 | 3.33 | 4.68 |
| NegGrad | 26.07 | 1.21 | 0.56 | 0.49 | 36.00 | 10.58 | 5.27 | 0.76 | 0.14 | 0.12 | 12.00 | 13.86 |
| CF-k | 22.67 | 1.55 | 0.00 | 0.00 | 0.00 | 0.00 | 4.67 | 0.70 | 0.00 | 0.00 | 0.00 | 0.00 |
| EU-k | 25.87 | 0.64 | 3.23 | 1.69 | 8.00 | 6.93 | 5.20 | 0.20 | 0.00 | 0.00 | 0.00 | 0.00 |
| Bad-T | 29.27 | 0.83 | 1.05 | 1.05 | 0.0 | 0.0 | 9.5 | 1.84 | 0.95 | 0.74 | 0.0 | 0.0 |
| SCRUBS | 23.88 | 1.78 | 0.08 | 0.12 | 40.8 | 8.2 | 3.87 | 0.72 | 0.0 | 0.0 | 25.33 | 4.13 |

Table 5: **Small-scale** results with All-CNN. SCRUBS is the top-performer in terms of forgetting with minimal performance degradation.

| Model | CIFAR-10 | | | | | | Lacuna-10 | | | | | |
|---|---|---|---|---|---|---|---|---|---|---|---|---|
| | Test error (↓) | | Retain error (↓) | | Forget error (↑) | | Test error (↓) | | Retain error (↓) | | Forget error (↑) | |
| | mean | std | mean | std | mean | std | mean | std | mean | std | mean | std |
| Retrain | 14.72 | 0.16 | 0.0 | 0.0 | 100.0 | 0.0 | 2.87 | 0.34 | 0.0 | 0.0 | 99.75 | 0.56 |
| Original | 16.56 | 0.1 | 0.0 | 0.0 | 0.0 | 0.0 | 3.07 | 0.26 | 0.0 | 0.0 | 0.0 | 0.0 |
| Finetune | 16.41 | 0.09 | 0.0 | 0.0 | 0.0 | 0.0 | 3.02 | 0.37 | 0.0 | 0.0 | 0.0 | 0.0 |
| Fisher | 26.42 | 1.41 | 2.45 | 0.84 | 100.0 | 0.0 | 3.33 | 0.54 | 0.0 | 0.0 | 100.0 | 0.0 |
| NegGrad | 17.84 | 1.46 | 1.74 | 2.55 | 91.26 | 7.73 | 3.41 | 0.17 | 0.00 | 0.00 | 14.90 | 1.78 |
| CF-k | 15.31 | 0.12 | 0.00 | 0.00 | 0.03 | 0.01 | 2.89 | 0.22 | 0.00 | 0.00 | 0.00 | 0.00 |
| EU-k | 18.73 | 0.42 | 0.00 | 0.00 | 98.79 | 0.18 | 3.19 | 0.17 | 0.01 | 0.02 | 4.06 | 0.83 |
| SCRUBS | 15.73 | 0.17 | 0.51 | 0.02 | 100.0 | 0.0 | 3.69 | 0.36 | 0.28 | 0.23 | 100.0 | 0.0 |

Table 6: **Large-scale, class unlearning** results with ResNet. SCRUBS and EU-k are the top-performers in this setting in terms of forgetting with minimal performance degradation. Note, however, that EU-k doesn't perform strongly across the board (Figure 3) and in particular performs very poorly in selective unlearning (notice the contrast between EU-k's forget error between Figures 4a and 4b and between Figures 7b and 8b). Fisher is also a top-performer in terms of forget error in this setting too, but on CIFAR causes a large degradation in test error, as is often observed for this method.

| Model | CIFAR-10 | | | | | | Lacuna-10 | | | | | |
|---|---|---|---|---|---|---|---|---|---|---|---|---|
| | Test error (↓) | | Retain error (↓) | | Forget error (↑) | | Test error (↓) | | Retain error (↓) | | Forget error (↑) | |
| | mean | std | mean | std | mean | std | mean | std | mean | std | mean | std |
| Retrain | 13.97 | 0.19 | 0.0 | 0.0 | 100.0 | 0.0 | 1.59 | 0.36 | 0.0 | 0.0 | 100.0 | 0.0 |
| Original | 15.56 | 0.25 | 0.0 | 0.0 | 0.0 | 0.0 | 1.56 | 0.33 | 0.0 | 0.0 | 0.0 | 0.0 |
| Finetune | 15.39 | 0.22 | 0.0 | 0.0 | 0.0 | 0.0 | 1.67 | 0.44 | 0.0 | 0.0 | 0.0 | 0.0 |
| Fisher | 27.4 | 2.28 | 3.66 | 1.03 | 99.0 | 0.0 | 1.78 | 0.29 | 0.0 | 0.0 | 89.0 | 0.0 |
| NegGrad | 17.87 | 0.31 | 0.58 | 0.13 | 87.22 | 1.67 | 1.63 | 0.17 | 0.00 | 0.00 | 6.56 | 1.13 |
| CF-k | 14.99 | 0.23 | 0.00 | 0.00 | 0.00 | 0.00 | 1.48 | 0.36 | 0.00 | 0.00 | 0.00 | 0.00 |
| EU-k | 15.30 | 0.69 | 0.13 | 0.14 | 100.00 | 0.00 | 1.74 | 0.45 | 0.00 | 0.00 | 77.19 | 39.51 |
| SCRUBS | 15.06 | 0.14 | 0.12 | 0.03 | 100.0 | 0.0 | 2.0 | 0.4 | 0.0 | 0.0 | 100.0 | 0.0 |

Table 7: **Large-scale, class unlearning** results with All-CNN. SCRUBS is the top-performer in terms of forgetting with minimal performance degradation.

| Model | CIFAR-10 | | | | | | Lacuna-10 | | | | | |
| | Test error (↓) | | Retain error (↓) | | Forget error (↑) | | Test error (↓) | | Retain error (↓) | | Forget error (↑) | |
| | mean | std | mean | std | mean | std | mean | std | mean | std | mean | std |
|---|---|---|---|---|---|---|---|---|---|---|---|---|
| Retrain | 17.4 | 0.14 | 0.0 | 0.0 | 29.67 | 3.21 | 2.7 | 0.2 | 0.0 | 0.0 | 1.0 | 1.0 |
| Original | 17.36 | 0.14 | 0.0 | 0.0 | 0.0 | 0.0 | 2.73 | 0.15 | 0.0 | 0.0 | 0.0 | 0.0 |
| Finetune | 17.37 | 0.11 | 0.0 | 0.0 | 0.0 | 0.0 | 2.63 | 0.12 | 0.0 | 0.0 | 0.0 | 0.0 |
| Fisher | 21.23 | 0.27 | 2.88 | 0.54 | 3.0 | 2.65 | 3.1 | 0.35 | 0.0 | 0.0 | 0.0 | 0.0 |
| NegGrad | 22.7 | 0.6 | 4.1 | 0.5 | 53.7 | 6.8 | 4.7 | 0.2 | 0.9 | 0.1 | 13.0 | 1.0 |
| CF-k | 17.4 | 0.1 | 0.0 | 0.0 | 0.0 | 0.0 | 2.7 | 0.2 | 0.0 | 0.0 | 0.0 | 0.0 |
| EU-k | 21.8 | 0.2 | 0.4 | 0.6 | 23.7 | 3.5 | 2.9 | 0.1 | 0.0 | 0.0 | 0.0 | 0.0 |
| SCRUBS | 18.04 | 0.2 | 0.0 | 0.0 | 70.33 | 4.16 | 3.0 | 0.0 | 0.0 | 0.0 | 4.67 | 3.06 |

Table 8: **Large-scale, selective unlearning** results with ResNet. SCRUBS and NegGrad are the top-performers in terms of forgetting, though NegGrad has worse test performance than SCRUBS in both cases. Note also that NegGrad isn't as consistent at forgetting across settings as SCRUBS, as can be seen in Figure 3.

| Model | CIFAR-10 | | | | | | Lacuna-10 | | | | | |
| | Test error (↓) | | Retain error (↓) | | Forget error (↑) | | Test error (↓) | | Retain error (↓) | | Forget error (↑) | |
| | mean | std | mean | std | mean | std | mean | std | mean | std | mean | std |
|---|---|---|---|---|---|---|---|---|---|---|---|---|
| Retrain | 16.47 | 0.21 | 0.0 | 0.0 | 25.67 | 2.31 | 1.6 | 0.44 | 0.0 | 0.0 | 0.67 | 0.58 |
| Original | 16.43 | 0.08 | 0.0 | 0.0 | 0.0 | 0.0 | 1.53 | 0.31 | 0.0 | 0.0 | 0.0 | 0.0 |
| Finetune | 16.5 | 0.18 | 0.0 | 0.0 | 0.0 | 0.0 | 1.43 | 0.21 | 0.0 | 0.0 | 0.0 | 0.0 |
| Fisher | 21.39 | 1.22 | 4.0 | 1.44 | 13.0 | 11.27 | 1.87 | 0.21 | 0.01 | 0.02 | 0.0 | 0.0 |
| NegGrad | 21.36 | 0.34 | 3.23 | 0.37 | 45.33 | 2.89 | 2.77 | 0.25 | 0.40 | 0.07 | 8.67 | 0.58 |
| CF-k | 16.29 | 0.07 | 0.00 | 0.00 | 0.00 | 0.00 | 1.53 | 0.31 | 0.00 | 0.00 | 0.00 | 0.00 |
| EU-k | 17.62 | 0.61 | 0.11 | 0.11 | 0.33 | 0.58 | 1.83 | 0.47 | 0.00 | 0.00 | 0.00 | 0.00 |
| SCRUBS | 16.55 | 0.11 | 0.0 | 0.0 | 29.33 | 3.21 | 2.07 | 0.31 | 0.0 | 0.0 | 1.67 | 0.58 |

Table 9: **Large-scale, selective unlearning** results with All-CNN. SCRUBS and NegGrad are the top-performers in terms of forgetting, though NegGrad has worse test performance than SCRUBS in both cases. Note also that NegGrad isn't as consistent at forgetting across settings as SCRUBS, as can be seen in Figure 3.

| Model | Forget 3 classes (50%) | | | | | | Forget 4 classes (∼ 67%) | | | | | |
| | Test error (↓) | | Retain error (↓) | | Forget error (↑) | | Test error (↓) | | Retain error (↓) | | Forget error (↑) | |
| | mean | std | mean | std | mean | std | mean | std | mean | std | mean | std |
|---|---|---|---|---|---|---|---|---|---|---|---|---|
| retrain | **32.2** | 2.22 | 0.0 | 0.0 | **99.9** | 0.19 | **14.5** | 2.18 | 0.0 | 0.0 | **99.8** | 0.14 |
| Original | 37.7 | 3.0 | 0.0 | 0.0 | 0.0 | 0.0 | 33.7 | 1.04 | 0.0 | 0.0 | 0.0 | 0.0 |
| Finetune | **33.2** | 3.0 | 0.0 | 0.0 | 4.9 | 0.5 | 20.3 | 2.75 | 0.0 | 0.0 | 9.2 | 0.9 |
| Fisher | 37.4 | 2.7 | 15.4 | 7.6 | 84.4 | 6.4 | 21.8 | 7.09 | 7.83 | 9.09 | 84.3 | 2.8 |
| NTK | 35.2 | 2.9 | 0.0 | 0.0 | 25.7 | 22.3 | 18.5 | 3.8 | 0.0 | 0.0 | 46.9 | 5.9 |
| NegGrad | 31.1 | 2.2 | 1.2 | 1.5 | 89.2 | 1.8 | **12.8** | 1.8 | 0.0 | 0.0 | 78.8 | 1.4 |
| CF-k | **32.6** | 3.1 | 0.0 | 0.0 | 8.2 | 1.0 | 18.8 | 2.4 | 0.0 | 0.0 | 14.3 | 1.2 |
| EU-k | **32.6** | 3.9 | 0.2 | 0.2 | 55.4 | 2.2 | **15.7** | 1.8 | 0.2 | 0.3 | 64.3 | 0.7 |
| Bad-T | **32.7** | 2.4 | 0.22 | 0.19 | **99.7** | 0.6 | **14.5** | 2.6 | 0.0 | 0.0 | **97.1** | 2.6 |
| SCRUBS | **30.8** | 1.3 | 0.0 | 0.0 | **98.6** | 2.0 | **15.3** | 2.3 | 0.0 | 0.0 | **99.6** | 0.7 |

Table 10: For **large forget sets, class unlearning** on CIFAR-6 (with ResNet), SCRUBS achieves significantly better forgetting than all alternatives and doesn't degrade performance.

| | Forget 50% | | | | | | Forget ∼ 67% | | | | | |
|---|---|---|---|---|---|---|---|---|---|---|---|---|
| | Test error (↓) | | Retain error (↓) | | Forget error (↑) | | Test error (↓) | | Retain error (↓) | | Forget error (↑) | |
| Model | mean | std | mean | std | mean | std | mean | std | mean | std | mean | std |
| Retrain | 28.56 | 1.68 | 0.0 | 0.0 | 100.0 | 0.0 | 13.17 | 2.25 | 0.0 | 0.0 | 100.0 | 0.0 |
| Original | 34.22 | 1.84 | 0.0 | 0.0 | 5.67 | 9.81 | 30.33 | 9.28 | 1.17 | 1.61 | 4.08 | 5.57 |
| Finetune | 29.11 | 3.5 | 0.33 | 0.58 | 25.44 | 20.21 | 13.5 | 5.07 | 0.0 | 0.0 | 21.67 | 7.35 |
| Fisher | 42.33 | 4.51 | 19.89 | 4.34 | 75.67 | 13.13 | 27.67 | 13.34 | 12.67 | 5.35 | 80.42 | 14.05 |
| NTK | 31.56 | 1.17 | 0.0 | 0.0 | 27.11 | 8.95 | 25.33 | 5.25 | 0.83 | 1.04 | 24.33 | 9.96 |
| NegGrad | 29.78 | 4.03 | 2.44 | 2.04 | 100.0 | 0.0 | 11.0 | 2.78 | 0.33 | 0.58 | 100.0 | 0.0 |
| CF-k | 31.67 | 3.48 | 0.0 | 0.0 | 6.89 | 10.22 | 19.0 | 1.5 | 0.0 | 0.0 | 6.33 | 7.72 |
| EU-k | 33.67 | 1.67 | 5.0 | 0.33 | 100.0 | 0.0 | 13.33 | 2.84 | 1.67 | 1.15 | 100.0 | 0.0 |
| Bad-T | 31.22 | 4.79 | 0.67 | 0.88 | 98.33 | 0.67 | 15.33 | 2.89 | 0.67 | 0.76 | 87.33 | 5.11 |
| SCRUBS | 28.78 | 2.36 | 0.0 | 0.0 | 100.0 | 0.0 | 12.67 | 1.44 | 0.0 | 0.0 | 99.92 | 0.14 |

Table 11: **Large forget sets, class unlearning** on CIFAR-6 with All-CNN. SCRUBS, NegGrad and EU-k are the top-performers in this setup, though EU-k sometimes incurs performance degradation. Note that, generally, EU-k doesn't perform strongly across the board (Figure 3) and in particular performs very poorly in selective unlearning (notice the contrast between EU-k's forget error between Figures 4a and 4b and between Figures 7b and 8b). NegGrad also isn't nearly as consistent as SCRUBS in forgetting across different scenarios (Figure 3)

| | Forget 50% | | | | | | Forget ∼ 67% | | | | | |
|---|---|---|---|---|---|---|---|---|---|---|---|---|
| | Test error (↓) | | Retain error (↓) | | Forget error (↑) | | Test error (↓) | | Retain error (↓) | | Forget error (↑) | |
| Model | mean | std | mean | std | mean | std | mean | std | mean | std | mean | std |
| Retrain | 3.9 | 0.7 | 0.0 | 0.0 | 97.9 | 3.7 | 1.5 | 0.9 | 0.0 | 0.0 | 96.3 | 6.1 |
| Original | 6.1 | 1.3 | 0.2 | 0.2 | 0.1 | 0.2 | 6.0 | 1.7 | 0.3 | 0.3 | 0.1 | 0.1 |
| Finetune | 3.8 | 0.4 | 0.0 | 0.0 | 4.1 | 1.4 | 2.5 | 1.8 | 0.0 | 0.0 | 4.1 | 2.6 |
| Fisher | 7.4 | 1.9 | 3.4 | 0.8 | 95.6 | 4.0 | 4.3 | 1.3 | 1.8 | 1.2 | 95.3 | 3.5 |
| NTK | 5.0 | 0.9 | 0.0 | 0.0 | 6.0 | 0.9 | 4.5 | 1.3 | 0.0 | 0.0 | 7.9 | 2.5 |
| NegGrad | 3.9 | 0.7 | 0.0 | 0.0 | 64.7 | 9.5 | 2.7 | 1.3 | 0.0 | 0.0 | 40.5 | 4.8 |
| CF-k | 4.1 | 0.7 | 0.0 | 0.0 | 0.8 | 0.2 | 3.5 | 1.3 | 0.0 | 0.0 | 0.3 | 0.4 |
| EU-k | 3.3 | 0.6 | 0.0 | 0.0 | 24.7 | 6.2 | 1.5 | 0.5 | 0.0 | 0.0 | 20.8 | 4.9 |
| Bad-T | 4.44 | 0.19 | 2.0 | 0.33 | 99.89 | 0.19 | 1.17 | 0.76 | 1.17 | 0.76 | 91.08 | 4.14 |
| SCRUBS | 4.1 | 0.8 | 0.0 | 0.0 | 99.2 | 1.4 | 1.8 | 0.3 | 0.0 | 0.0 | 97.1 | 1.5 |

Table 12: **Large forget sets, class unlearning** on Lacuna-6 with ResNet. SCRUBS is the top-performer in terms of forgetting with minimal performance degradation.

| | Forget 50% | | | | | | Forget ∼ 67% | | | | | |
|---|---|---|---|---|---|---|---|---|---|---|---|---|
| | Test error (↓) | | Retain error (↓) | | Forget error (↑) | | Test error (↓) | | Retain error (↓) | | Forget error (↑) | |
| Model | mean | std | mean | std | mean | std | mean | std | mean | std | mean | std |
| Retrain | 2.89 | 0.77 | 0.0 | 0.0 | 100.0 | 0.0 | 0.17 | 0.29 | 0.0 | 0.0 | 100.0 | 0.0 |
| Original | 4.44 | 0.51 | 0.0 | 0.0 | 0.0 | 0.0 | 3.83 | 0.58 | 0.0 | 0.0 | 0.0 | 0.0 |
| Finetune | 2.44 | 0.19 | 0.0 | 0.0 | 10.22 | 1.02 | 4.17 | 5.97 | 2.83 | 4.91 | 36.0 | 50.77 |
| Fisher | 44.78 | 11.6 | 39.78 | 11.97 | 83.22 | 20.61 | 47.67 | 4.54 | 44.5 | 7.47 | 84.75 | 16.45 |
| NTK | 3.56 | 0.38 | 0.0 | 0.0 | 6.44 | 0.51 | 2.83 | 0.76 | 0.0 | 0.0 | 7.17 | 3.19 |
| NegGrad | 2.33 | 0.33 | 0.0 | 0.0 | 100.0 | 0.0 | 0.33 | 0.29 | 0.0 | 0.0 | 100.0 | 0.0 |
| CF-k | 3.0 | 0.58 | 0.0 | 0.0 | 0.22 | 0.19 | 2.17 | 0.29 | 0.0 | 0.0 | 0.0 | 0.0 |
| EU-k | 3.33 | 0.58 | 0.0 | 0.0 | 89.22 | 18.67 | 0.17 | 0.29 | 0.0 | 0.0 | 84.5 | 26.85 |
| Bad-T | 5.33 | 1.2 | 1.11 | 0.69 | 100.0 | 0.0 | 1.17 | 0.29 | 0.83 | 0.76 | 99.25 | 0.43 |
| SCRUBS | 2.56 | 0.77 | 0.0 | 0.0 | 99.44 | 0.96 | 1.0 | 1.0 | 0.0 | 0.0 | 100.0 | 0.0 |

Table 13: **Large forget sets, class unlearning** on Lacuna-6 with All-CNN. SCRUBS and EU-k are the top-performers in this setting in terms of forgetting with minimal performance degradation. But note that, generally, EU-k doesn't perform strongly across the board (Figure 3) and in particular performs very poorly in selective unlearning (notice the contrast between EU-k's forget error between Figures 4a and 4b and between Figures 7b and 8b).

| Model | CIFAR-6 | | | | | | Lacuna-6 | | | | | |
|---|---|---|---|---|---|---|---|---|---|---|---|---|
| | Test error (↓) | | Retain error (↓) | | Forget error (↑) | | Test error (↓) | | Retain error (↓) | | Forget error (↑) | |
| | mean | std | mean | std | mean | std | mean | std | mean | std | mean | std |
| Retrain | 35.06 | 1.02 | 0.0 | 0.0 | 38.22 | 2.5 | 6.56 | 0.35 | 0.0 | 0.0 | 6.11 | 0.19 |
| Original | 32.06 | 0.69 | 0.0 | 0.0 | 0.0 | 0.0 | 5.5 | 0.73 | 0.0 | 0.0 | 0.0 | 0.0 |
| Finetune | 32.11 | 1.17 | 0.0 | 0.0 | 0.0 | 0.0 | 5.72 | 0.54 | 0.0 | 0.0 | 0.0 | 0.0 |
| Fisher | 38.06 | 1.95 | 14.44 | 4.83 | 17.56 | 7.31 | 14.44 | 4.11 | 7.0 | 2.31 | 6.89 | 1.39 |
| NTK | 33.28 | 0.69 | 0.0 | 0.0 | 27.33 | 4.26 | 6.17 | 0.67 | 0.0 | 0.0 | 5.22 | 0.38 |
| NegGrad | 32.0 | 1.09 | 0.0 | 0.0 | 0.0 | 0.0 | 5.67 | 0.58 | 0.0 | 0.0 | 0.0 | 0.0 |
| CF-k | 32.0 | 1.32 | 0.0 | 0.0 | 0.0 | 0.0 | 5.72 | 0.67 | 0.0 | 0.0 | 0.0 | 0.0 |
| EU-k | 34.11 | 0.96 | 1.11 | 0.38 | 15.78 | 2.27 | 5.56 | 0.51 | 0.11 | 0.19 | 0.89 | 0.69 |
| Bad-T | 37.33 | 2.75 | 2.67 | 0.58 | 23.56 | 3.67 | 9.33 | 0.33 | 0.0 | 0.0 | 5.56 | 1.02 |
| SCRUBS | 33.89 | 2.58 | 0.11 | 0.19 | 23.11 | 2.83 | 6.22 | 0.1 | 0.0 | 0.0 | 7.33 | 0.88 |

Table 14: **Large forget sets, selective unlearning** with ResNet. In this setting, SCRUBS and NTK are top-performers on forgetting without sacrificing model performance. While Fisher also achieves forgetting to some extent, notice that it substantially degrades performance. On the other hand, NTK in general suffers from poor unlearning performance in most other settings considered (e.g. see Figures 1, 2 and 3) and it additionally fails to scale to datasets of the size of CIFAR-10 / Lacuna-10.

| Model | CIFAR-6 | | | | | | Lacuna-6 | | | | | |
|---|---|---|---|---|---|---|---|---|---|---|---|---|
| | Test error (↓) | | Retain error (↓) | | Forget error (↑) | | Test error (↓) | | Retain error (↓) | | Forget error (↑) | |
| | mean | std | mean | std | mean | std | mean | std | mean | std | mean | std |
| Retrain | 35.11 | 4.25 | 1.11 | 1.92 | 35.0 | 2.65 | 5.22 | 0.42 | 0.0 | 0.0 | 4.11 | 2.5 |
| Original | 31.17 | 5.83 | 2.89 | 5.0 | 2.78 | 4.81 | 4.5 | 0.44 | 0.11 | 0.19 | 0.0 | 0.0 |
| Finetune | 30.33 | 3.06 | 0.0 | 0.0 | 0.0 | 0.0 | 4.78 | 1.93 | 0 | 0.0 | 0.0 | 0.0 |
| Fisher | 46.28 | 5.62 | 30.78 | 12.54 | 31.56 | 8.77 | 52.06 | 12.86 | 46.0 | 11.22 | 51.33 | 13.12 |
| NTK | 31.83 | 5.45 | 2.56 | 4.43 | 16.0 | 2.85 | 4.89 | 0.25 | 0.0 | 0.0 | 3.67 | 2.08 |
| NegGrad | 29.94 | 3.67 | 0.0 | 0.0 | 0.11 | 0.19 | 4.28 | 0.54 | 0.11 | 0.19 | 0.78 | 0.84 |
| CF-k | 30.0 | 3.03 | 0.11 | 0.19 | 0.0 | 0.0 | 4.5 | 0.6 | 0.0 | 0.0 | 0.0 | 0.0 |
| EU-k | 41.22 | 0.77 | 16.22 | 2.69 | 18.89 | 2.01 | 5.83 | 0.73 | 0.11 | 0.19 | 0.44 | 0.38 |
| Bad-T | 39.06 | 2.7 | 0.22 | 0.19 | 35.56 | 3.91 | 6.33 | 1.26 | 0.0 | 0.0 | 3.44 | 1.95 |
| SCRUBS | 30.83 | 2.91 | 0.0 | 0.0 | 12.44 | 1.26 | 4.61 | 1.07 | 0.0 | 0.0 | 5.89 | 0.84 |

Table 15: **Large forget sets, selective unlearning** with All-CNN. Fisher is the top-performer in terms of forget set error, but it introduces large degradation in both test and retain performance. EU-k also performs well on forgetting (though only on one of the 2 datasets) but also degrades performance. SCRUBS strikes a good balance between forgetting while retaining performance.

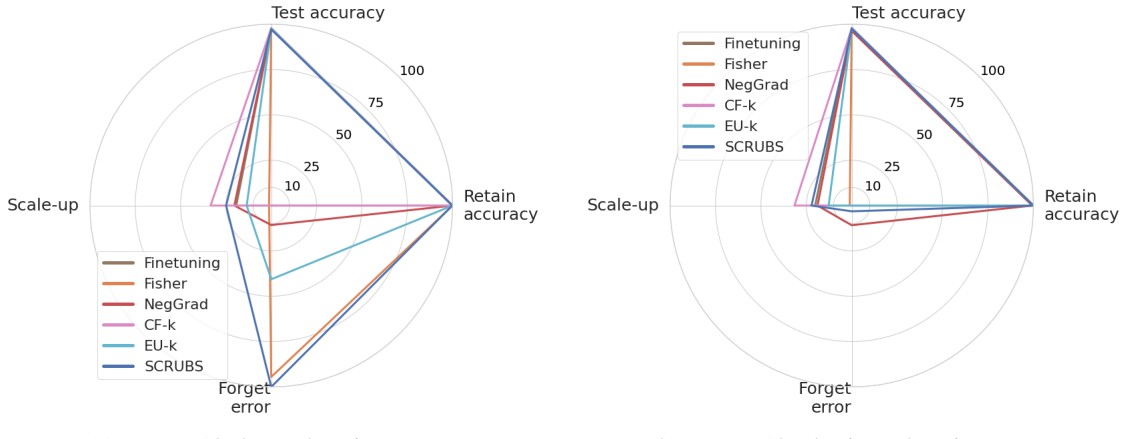

(a) Lacuna-10 class unlearning.      (b) Lacuna-10 selective unlearning.

Figure 6: In the **large-scale setting** on Lacuna-10, SCRUBS is among the top-performers: it is slightly slower than CF-k, but CF-k doesn't achieve unlearning (its forget error is 0%) and it is slightly outperformed by NegGrad on the selective setting, but NegGrad is not consistently strong-performing and, in fact performs very poorly on class unlearning. Interestingly, selective forgetting is significantly harder for all methods, but especially for EU-k and Fisher which completely fail to unlearn in the selective case. Scale-up is computed as the fraction of runtime of the retrain from scratch baseline over the runtime of the given unlearning algorithm. This fraction is multiplied by 10 in this plot for visualization purposes. Each point represents the average across ResNet and All-CNN variants. For reference, the data used to create this plot lives in Tables 6 and 7 (for the left subplot) and Tables 8 and 9 (for the right subplot).

## A.5 Additional figures

In this section, we include additional figures that didn't fit in the main paper.

Specifically, we report the large-scale spider plot for Lacuna-10 in Figure 6. We also separate out the settings of class- and selective- unlearning for large forget set sizes, in Figures 7 and 8. These two complement Figure 1 in the main paper which aggregates over both class- and selective- unlearning. Separating them out allows to observe different trends. Selective unlearning is substantially harder for all methods when it comes to forgetting. Uncharacteristically, NTK and Fisher are among the top performers in that regard when it comes to selective unlearning. Fisher, though, causes a significant performance degradation, isn't generally a top performer in most settings we considered, and is very slow to run, making it impractical (Figure 4). NTK also generally suffers from poor unlearning performance in most other settings considered (e.g. see Figures 1, 2 and 3) and it additionally fails to scale to datasets of the size of CIFAR-10 / Lacuna-10. Regardless, these results reveal the difference in behaviours across the diverse set of settings we consider, and we hope consistutes a source of inspiration for future investigations.

In addition, we include additional plots of SCRUBS' training dynamics, for different datasets and architectures, in Figures 9, 10 and 11, where the conclusions are in agreement with our discussion of this topic in the main paper. Finally, we ablate the effect of cross-entropy in Figure 12 where we find that using cross-entropy offers an additional protection to maintaining the model's performance during the unlearning procedure. This sometimes comes at the cost of smaller forget set error, compared to the forget set error that would have been achieved if cross-entropy was omitted from the loss.

## A.6 Additional evaluation metrics: Resolving class confusion via unlearning

In this section, we explore the performance of SCRUBS and various other baselines on a different set of metrics for measuring forgetting, inspired by Goel et al. (2022). Specifically, we explore a scenario where the forget set has confused labels (e.g. for two classes A and B, examples of A are labelled as B, and vice versa). The idea here is that, because mislabelled examples are only present

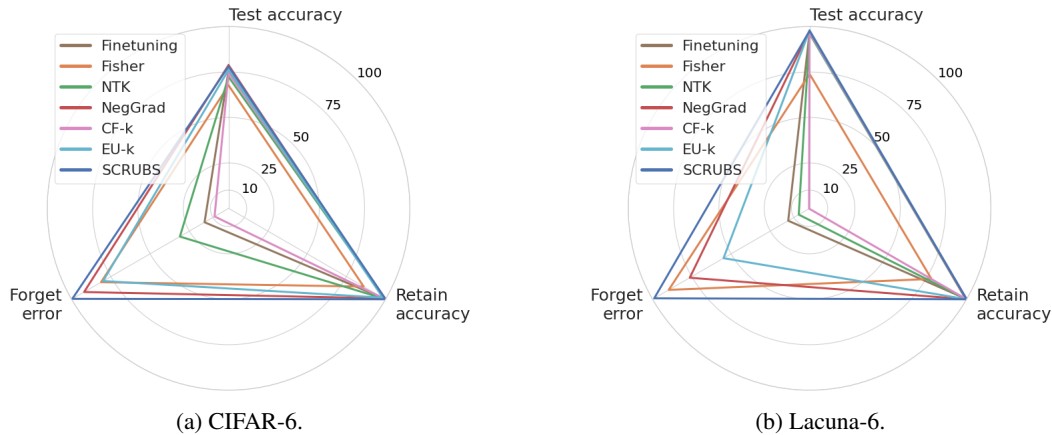

(a) CIFAR-6.

(b) Lacuna-6.

Figure 7: **Large forget sets, class unlearning**. SCRUBS is the only top-performer across settings. Each point represents the average across 4 settings: {ResNet, All-CNN} x {50%, 67%}. For reference, the data used to create these plots lives in Tables 1, 11, 12, 13.

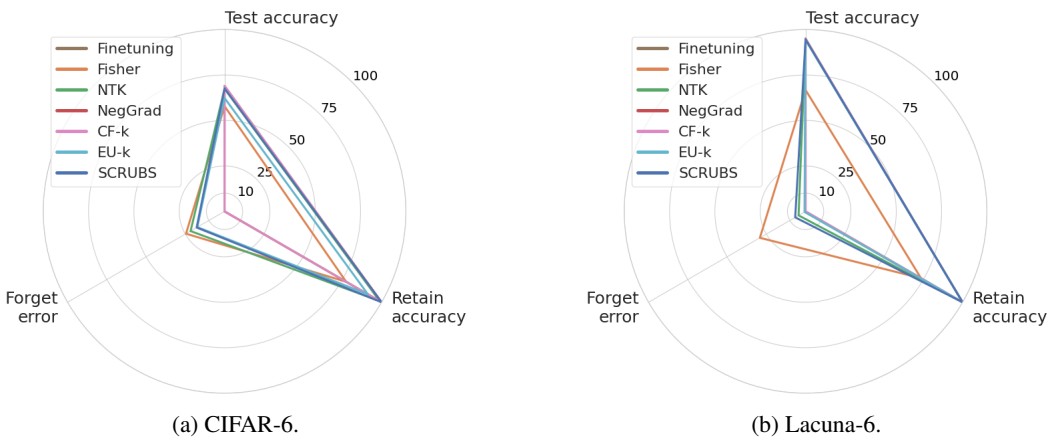

(a) CIFAR-6.

(b) Lacuna-6.

Figure 8: **Large forget sets, selective unlearning**. Each point represents the average across ResNet and All-CNN variants. On the left, NegGrad, Finetuning and CF-k perform very similarly, so their lines cover one-another and, on the right, the same is true for NegGrad, Finetuning, CF-k and EU-k. The detailed results can be seen in Tables 14 and 15. **Take-away**: SCRUBS remains a top performer in terms of being one of the best methods in terms of unlearning, with minimal performance degradation. Interestingly, we observe that NTK and Fisher perform uncharacteristically well here in terms of forgetting, contrary to the their performance in most other scenarios considered (see e.g. Figures 2, 4). Fisher, though, causes a significant performance degradation, isn't generally a top performer in most settings we considered, and is very slow to run, making it impractical (Figure 4). NTK also generally suffers from poor unlearning performance in most other settings considered (e.g. see Figures 1, 2 and 3) and it additionally fails to scale to datasets of the size of CIFAR-10 / Lacuna-10. Regardless, these results reveal the difference in behaviours across the diverse set of settings we consider, and we hope consistutes a source of inspiration for future investigations. For reference, the data used to create this plot lives in Tables 14 and 15.

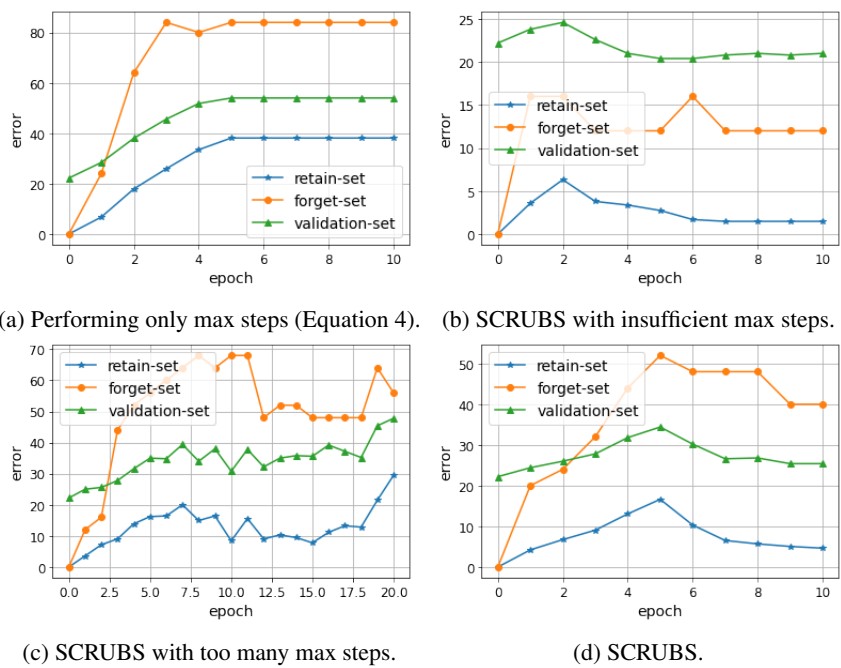

(a) Performing only max steps (Equation 4).    (b) SCRUBS with insufficient max steps.

(c) SCRUBS with too many max steps.        (d) SCRUBS.

Figure 9: Illustration of training dynamics of SCRUBS variants, on CIFAR-5 with All-CNN. Performing the right interleaving of *min-steps* and *max-steps* is important for achieving a good balance between high forget error and low retain and validation errors.

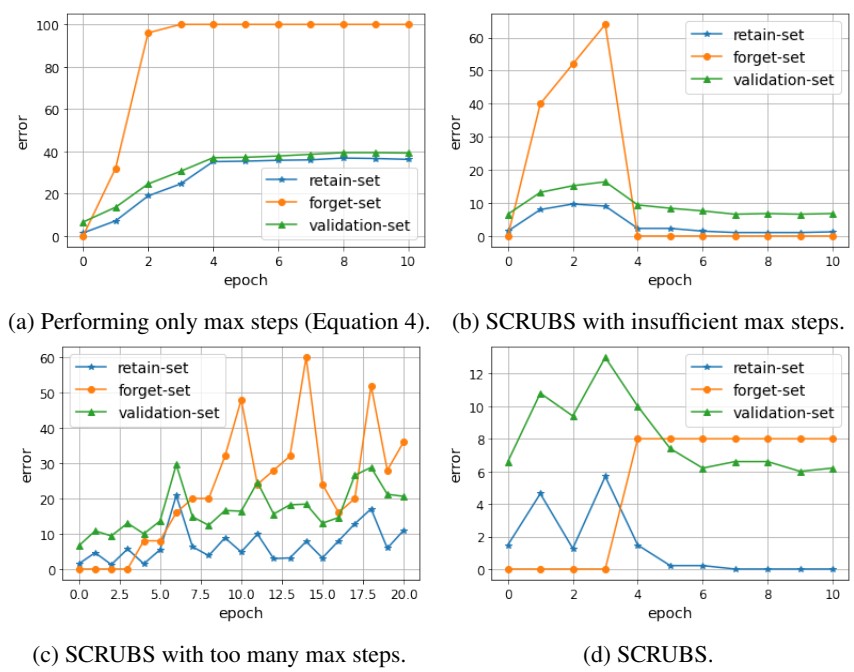

(a) Performing only max steps (Equation 4).    (b) SCRUBS with insufficient max steps.

(c) SCRUBS with too many max steps.        (d) SCRUBS.

Figure 10: Illustration of training dynamics of SCRUBS variants, on Lacuna-5 with ResNet. Performing the right interleaving of *min-steps* and *max-steps* is important for achieving a good balance between high forget error and low retain and validation errors.

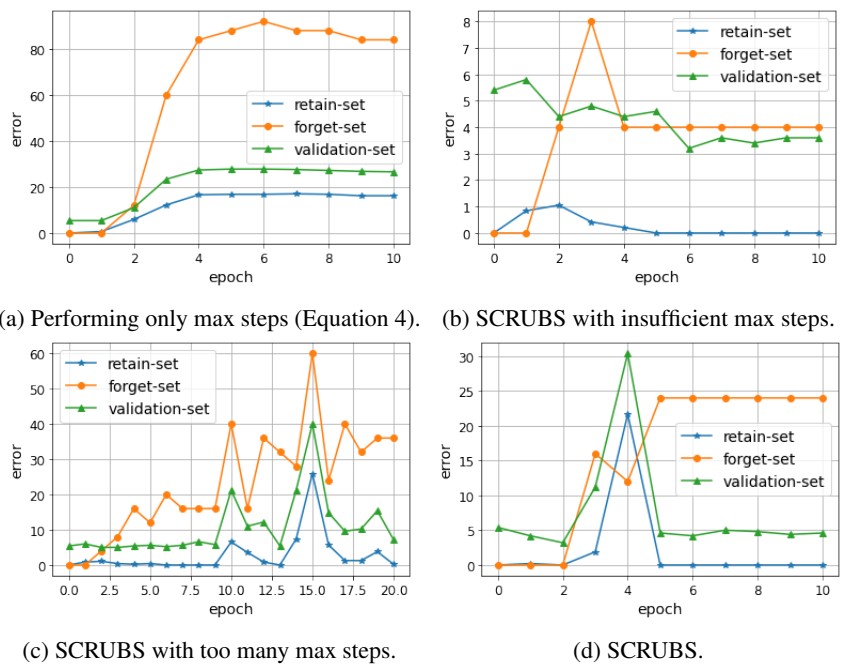

(a) Performing only max steps (Equation 4).    (b) SCRUBS with insufficient max steps.

(c) SCRUBS with too many max steps.       (d) SCRUBS.

Figure 11: Illustration of training dynamics of SCRUBS variants, on Lacuna-5 with All-CNN. Performing the right interleaving of *min-steps* and *max-steps* is important for achieving a good balance between high forget error and low retain and validation errors.

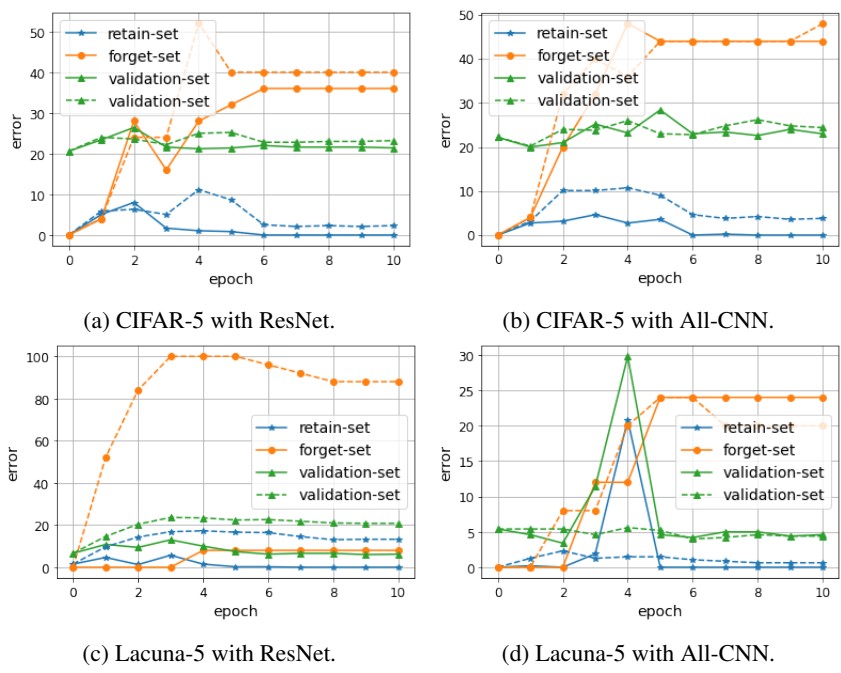

(a) CIFAR-5 with ResNet.         (b) CIFAR-5 with All-CNN.

(c) Lacuna-5 with ResNet.        (d) Lacuna-5 with All-CNN.

Figure 12: Effect of adding the cross-entropy loss in Equation 6. Dashed lines omit cross-entropy while solid lines use it. We find that the addition of cross-entropy offers an additional protection to maintaining the model's performance during the unlearning procedure. This sometimes comes at the cost of smaller forget set error, compared to the forget set error that would have been achieved if cross-entropy was omitted from the loss.

in the forget set, successful unlearning (removing the influence of the forget set) would lead to a model that is not at all confused between classes A and B.

In more detail, the setup we follow is: 1) We first mislabel some portion of the training dataset (we mislabelled examples between classes 0 and 1 of each of CIFAR-5 and Lacuna-5 in our experiments), 2) train the 'original model' on the (partly mislabelled) training dataset (it has mislabelled examples for classes 0 and 1 but correct labels for the remaining classes), 3) perform unlearning where the forget set contains all and only the confused examples. Given this, the goal for the unlearning algorithm is to resolve the confusion of the original model.

We consider the following metrics (using terminology consistent with Goel et al. (2022)). They are presented in order of decreasing generality, and increasing focus on measuring degrees of confusion between the two classes considered.

- **Error** (e.g. test error, retain error, forget error). This counts all mistakes, so anytime that an example of some class is predicted to be in any other class, it will be counted. These are the same metrics that we use for the rest of the paper (see Equation 7). For test and retain error, lower is better, whereas for forget error, higher is better.

- **Interclass Confusion IC-ERR** (e.g. IC test error, IC retain error). This counts only mistakes that involve examples from the confused classes A and B. Specifically, it counts instances of any example of class A being predicted to be in *any* other class , and analogously for class B. Compared to Error, this metric is more focused towards understanding the result of the introduced confusion, since it only considers cases that relate to the confused classes. A successful unlearning method would make no such errors, so lower is better for each of IC test error and IC retain error.

- **FGT-ERR** (e.g. Fgt test error, Fgt retain error). This metric counts only misclassification *between the confused classes* A and B. Here, a mistake of an example of class A (or B) being predicted to be in class other than A or B will not be counted. Only mistakes of an example of class A being predicted to be in class B, and vice versa, are counted. **This is the most focused metric that explicitly measures the amount of confusion remaining after unlearning**. A successful unlearning method would make no such errors, so lower is better for each of Fgt test and Fgt retain.

More formally, Error is the same as defined in Equation 7. Let us now mathematically define IC-ERR and FGT-ERR. We denote by $C^{w,\mathcal{D}}$ the confusion matrix for model parameterized by $w$ on the dataset $\mathcal{D}$, and let $\mathcal{D}_A$ denote the part of the dataset $\mathcal{D}$ that belongs to class $A$. So, for example $\mathcal{D}_{r_A}$ denotes the part of the retain set $\mathcal{D}_r$ that belongs to class $A$, and the entry $C^{w,\mathcal{D}}_{A,B}$ of the confusion matrix stores the number of times that a sample belonging to class $A$ was (mis)classified as belonging to class $B$ in the dataset $\mathcal{D}$ by the model parameterized by $w$. Then, we have:

$$\text{IC-ERR}(\mathcal{D}, A, B; w) = \frac{\sum_k C^{w,\mathcal{D}}_{A,k} + \sum_{k'} C^{w,\mathcal{D}}_{B,k'}}{|\mathcal{D}_A| + |\mathcal{D}_B|} \tag{9}$$

where $k \neq A, k' \neq B$.

So, for example, the 'IC test error' column in our tables is computed via IC-ERR$(\mathcal{D}_t, 0, 1; w)$, where $\mathcal{D}_t$ denotes the test set, and 0 and 1 are the two classes confused in our experiments. Analogously, 'IC retain error' is computed as IC-ERR$(\mathcal{D}_r, 0, 1; w)$

Finally:

$$\text{FGT-ERR}(\mathcal{D}, A, B; w) = C^{w,\mathcal{D}}_{A,B} + C^{w,\mathcal{D}}_{B,A} \tag{10}$$

That is, FGT-ERR only measures the misclassification between the two confused classes A and B. So, for example, the 'Fgt test error' in our tables is computed as FGT-ERR$(\mathcal{D}_t, 0, 1; w)$ and analogously 'Fgt retain error' is computed as FGT-ERR$(\mathcal{D}_r, 0, 1; w)$.

The results are shown in Tables 16, 17, 18 and 19 for each of ResNet and All-CNN on both CIFAR-5 and Lacuna-5. We observe that across the board, SCRUBS is a top-performer on this metric too (see the captions of the individual tables for more details about performance profile).

| model | Test error (↓) mean | std | Retain error (↓) mean | std | Forget error (↑) mean | std | IC test error (↓) mean | std | IC retain error (↓) mean | std | Fgt test error (↓) mean | std | Fgt retain error (↓) mean | std |
|---|---|---|---|---|---|---|---|---|---|---|---|---|---|---|
| Retrain | 26.67 | 2.87 | 0.0 | 0.0 | 90.33 | 1.53 | 24.0 | 1.8 | 0.0 | 0.0 | 18.33 | 4.16 | 0.0 | 0.0 |
| Original | 41.0 | 2.09 | 0.0 | 0.0 | 0.0 | 0.0 | 56.0 | 3.04 | 0.0 | 0.0 | 92.0 | 7.94 | 0.0 | 0.0 |
| Finetune | 38.13 | 1.42 | 0.0 | 0.0 | 0.0 | 0.0 | 52.0 | 3.12 | 0.0 | 0.0 | 79.33 | 10.07 | 0.0 | 0.0 |
| NegGrad | 36.27 | 0.42 | 0.0 | 0.0 | 12.67 | 21.94 | 47.5 | 5.27 | 0.0 | 0.0 | 69.0 | 13.53 | 0.0 | 0.0 |
| CF-k | 39.6 | 1.64 | 0.0 | 0.0 | 0.0 | 0.0 | 54.83 | 2.02 | 0.0 | 0.0 | 85.33 | 7.02 | 0.0 | 0.0 |
| EU-k | 37.47 | 1.62 | 7.33 | 1.26 | 43.67 | 2.08 | 47.0 | 4.77 | 8.33 | 4.73 | 63.33 | 9.71 | 3.67 | 2.52 |
| Fisher | 44.8 | 2.36 | 21.33 | 3.45 | 32.0 | 11.53 | 51.5 | 7.47 | 26.33 | 9.5 | 79.0 | 3.61 | 20.0 | 7.94 |
| NTK | 32.6 | 2.51 | 0.0 | 0.0 | 60.33 | 0.58 | 37.5 | 4.0 | 0.0 | 0.0 | 52.0 | 10.58 | 0.0 | 0.0 |
| SCRUBS | 25.93 | 3.13 | 1.08 | 0.52 | 96.0 | 1.73 | 19.0 | 3.91 | 0.0 | 0.0 | 19.67 | 7.51 | 0.0 | 0.0 |

Table 16: Confusion metrics on CIFAR-5 with ResNet. (Confused class 0,1; 50-50 samples). SCRUBS is the best-performer by far in terms of eliminating the confusion via unlearning (see the IC error and Fgt error columns), while not hurting performance for other classes (see e.g. the usual Error metrics in the first 3 groups of columns).

| model | Test error (↓) mean | std | Retain error (↓) mean | std | Forget error (↑) mean | std | IC test error (↓) mean | std | IC retain error (↓) mean | std | Fgt test error (↓) mean | std | Fgt retain error (↓) mean | std |
|---|---|---|---|---|---|---|---|---|---|---|---|---|---|---|
| Retrain | 24.4 | 2.75 | 0.0 | 0.0 | 90.67 | 4.04 | 19.0 | 1.32 | 0.0 | 0.0 | 11.33 | 4.62 | 0.0 | 0.0 |
| Original | 37.07 | 4.67 | 1.5 | 2.6 | 5.67 | 9.81 | 49.0 | 4.77 | 6.0 | 10.39 | 80.67 | 12.58 | 6.0 | 10.39 |
| Finetune | 34.33 | 3.35 | 0.0 | 0.0 | 3.0 | 5.2 | 43.67 | 7.29 | 0.0 | 0.0 | 67.33 | 16.04 | 0.0 | 0.0 |
| NegGrad | 33.53 | 4.47 | 0.0 | 0.0 | 13.33 | 21.36 | 42.33 | 11.34 | 0.0 | 0.0 | 62.0 | 22.65 | 0.0 | 0.0 |
| CF-k | 36.13 | 4.21 | 0.0 | 0.0 | 0.33 | 0.58 | 47.83 | 5.8 | 0.0 | 0.0 | 76.33 | 14.43 | 0.0 | 0.0 |
| EU-k | 51.6 | 1.0 | 27.67 | 3.5 | 52.67 | 6.03 | 59.5 | 5.22 | 38.33 | 6.66 | 68.67 | 15.57 | 19.67 | 10.41 |
| Fisher | 51.93 | 2.95 | 35.17 | 3.92 | 31.0 | 11.53 | 56.83 | 8.69 | 31.67 | 14.01 | 78.33 | 15.53 | 17.67 | 11.5 |
| NTK | 32.2 | 2.84 | 0.75 | 1.3 | 43.33 | 14.15 | 36.67 | 4.07 | 3.0 | 5.2 | 54.33 | 9.02 | 3.0 | 5.2 |
| SCRUBS | 25.0 | 3.14 | 0.0 | 0.0 | 93.33 | 2.52 | 26.0 | 4.44 | 0.0 | 0.0 | 18.0 | 11.14 | 0.0 | 0.0 |

Table 17: Confusion metrics on CIFAR-5 with All-CNN. (Confused class 0,1; 50-50 samples). SCRUBS is the best-performer by far in terms of eliminating the confusion via unlearning (see the IC error and Fgt error columns), while not hurting performance for other classes (see e.g. the usual Error metrics in the first 3 groups of columns).

| model | Test error (↓) mean | std | Retain error (↓) mean | std | Forget error (↑) mean | std | IC test error (↓) mean | std | IC retain error (↓) mean | std | Fgt test error (↓) mean | std | Fgt retain error (↓) mean | std |
|---|---|---|---|---|---|---|---|---|---|---|---|---|---|---|
| Retrain | 6.0 | 0.2 | 0.0 | 0.0 | 99.67 | 0.58 | 7.17 | 2.57 | 0.0 | 0.0 | 0.0 | 0.0 | 0.0 | 0.0 |
| Original | 27.07 | 3.33 | 1.67 | 0.88 | 4.33 | 1.53 | 57.5 | 6.26 | 6.67 | 3.51 | 108.0 | 14.18 | 6.67 | 3.51 |
| Finetune | 18.8 | 4.26 | 0.0 | 0.0 | 14.67 | 6.03 | 37.67 | 11.15 | 0.0 | 0.0 | 63.67 | 22.01 | 0.0 | 0.0 |
| NegGrad | 17.8 | 2.95 | 1.67 | 0.72 | 55.33 | 2.08 | 33.17 | 5.25 | 5.33 | 1.53 | 56.67 | 12.9 | 4.33 | 1.53 |
| CF-k | 22.27 | 4.31 | 0.08 | 0.14 | 10.67 | 5.03 | 46.33 | 10.97 | 0.33 | 0.58 | 81.67 | 23.01 | 0.33 | 0.58 |
| EU-k | 15.27 | 3.19 | 0.83 | 0.38 | 62.0 | 12.49 | 29.33 | 9.0 | 2.33 | 1.53 | 43.67 | 16.29 | 0.33 | 0.58 |
| Fisher | 35.87 | 3.33 | 17.75 | 3.78 | 27.33 | 3.79 | 60.0 | 5.27 | 31.0 | 7.94 | 109.0 | 14.53 | 30.0 | 7.0 |
| NTK | 14.53 | 5.22 | 0.0 | 0.0 | 51.67 | 23.18 | 27.17 | 11.3 | 0.0 | 0.0 | 43.33 | 25.32 | 0.0 | 0.0 |
| SCRUBS | 8.47 | 1.17 | 0.33 | 0.14 | 96.0 | 1.0 | 11.33 | 3.82 | 1.33 | 0.58 | 9.33 | 1.53 | 1.33 | 0.58 |

Table 18: Confusion metrics on Lacuna-5 with ResNet. (Confused class 0,1; 50-50 samples). SCRUBS is the best-performer by far in terms of eliminating the confusion via unlearning (see the IC error and Fgt error columns), while not hurting performance for other classes (see e.g. the usual Error metrics in the first 3 groups of columns). NTK in some cases is able to resolve confusion, but not consistently, and it also suffers from higher Test Error.

| model | Test error (↓) mean | std | Retain error (↓) mean | std | Forget error (↑) mean | std | IC test error (↓) mean | std | IC retain error (↓) mean | std | Fgt test error (↓) mean | std | Fgt retain error (↓) mean | std |
|---|---|---|---|---|---|---|---|---|---|---|---|---|---|---|
| Retrain | 4.2 | 0.87 | 0.0 | 0.0 | 100.0 | 0.0 | 5.33 | 2.25 | 0.0 | 0.0 | 0.0 | 0.0 | 0.0 | 0.0 |
| Original | 25.47 | 2.32 | 5.75 | 5.63 | 20.33 | 25.74 | 56.17 | 4.93 | 23.0 | 22.54 | 105.67 | 8.08 | 23.0 | 22.54 |
| Finetune | 12.8 | 2.8 | 0.0 | 0.0 | 23.0 | 7.94 | 25.83 | 7.75 | 0.0 | 0.0 | 39.67 | 12.74 | 0.0 | 0.0 |
| NegGrad | 12.8 | 9.06 | 2.5 | 3.12 | 90.0 | 6.56 | 20.33 | 17.04 | 5.0 | 6.24 | 12.67 | 11.68 | 2.67 | 3.79 |
| CF-k | 21.27 | 1.63 | 0.58 | 0.8 | 9.33 | 0.58 | 47.0 | 4.58 | 2.33 | 3.21 | 82.67 | 10.12 | 2.33 | 3.21 |
| EU-k | 17.0 | 8.91 | 3.92 | 3.99 | 92.33 | 4.93 | 35.0 | 18.26 | 13.0 | 11.36 | 3.67 | 4.73 | 0.0 | 0.0 |
| Fisher | 49.6 | 4.73 | 39.25 | 7.45 | 40.0 | 9.54 | 57.67 | 10.79 | 42.33 | 11.59 | 88.67 | 11.68 | 29.67 | 16.86 |
| NTK | 12.87 | 6.63 | 2.83 | 4.91 | 72.33 | 12.06 | 25.5 | 17.88 | 11.33 | 19.63 | 35.67 | 24.03 | 10.0 | 17.32 |
| SCRUBS | 3.87 | 0.7 | 0.0 | 0.0 | 100.0 | 0.0 | 4.33 | 1.26 | 0.0 | 0.0 | 0.0 | 0.0 | 0.0 | 0.0 |

Table 19: Confusion metrics on Lacuna-5 with All-CNN. (Confused class 0,1; 50-50 samples). SCRUBS is the best-performer by far in terms of eliminating the confusion via unlearning (see the IC error and Fgt error columns), while not hurting performance for other classes (see e.g. the usual Error metrics in the first 3 groups of columns).

| $\beta$ | Test error | | Forget error | | Retain error | |
|---|---|---|---|---|---|---|
| | mean | std | mean | std | mean | std |
| 0.77 | 100.0 | 0.0 | 66.67 | 0.0 | 100.0 | 0.0 |
| 0.79 | 100.0 | 0.0 | 66.67 | 0.0 | 100.0 | 0.0 |
| 0.80 | 56.2 | 18.0 | 100.0 | 0.0 | 50.6 | 27.7 |
| 0.81 | 62.4 | 7.3 | 100.0 | 0.0 | 60.8 | 10.0 |
| 0.83 | 61.3 | 5.0 | 100.0 | 0.0 | 57.0 | 8.7 |
| 0.85 | 56.8 | 8.3 | 100.0 | 0.0 | 51.3 | 14.9 |
| 0.89 | 37.3 | 4.7 | 90.7 | 13.3 | 14.3 | 10.7 |
| 0.91 | 38.4 | 6.7 | 86.0 | 23.1 | 19.6 | 16.7 |
| 0.93 | 34.2 | 2.3 | 63.2 | 16.6 | 1.4 | 1.3 |
| 0.95 | 31.1 | 2.2 | 89.2 | 1.8 | 1.2 | 1.5 |
| 0.97 | 35.0 | 1.5 | 20.9 | 7.5 | 0.0 | 0.0 |
| 0.99 | 36.8 | 2.2 | 8.1 | 1.5 | 0.0 | 0.0 |

Table 20: NegGrad trade-off between forgetting and retaining. We observe that smaller $\beta$ tends to forget more and larger $\beta$ tends to overfit more to the retain-set based on Equation 8. The results correspond to the experiment in Table 1, for the experiment of large forget set, class unlearning on CIFAR-6 (forget 3 classes, i.e. 50%), with ResNet.

### A.7 TRADE-OFFS THAT ARISE WHEN TUNING MODELS

In this section, we discuss trade-offs present when tuning models. We begin with a case study of the NegGrad approach, which is the second strongest model overall, after SCRUBS. This will allow to better understand its performance profile under different settings of the hyperparameters. As shown in Equation 8, this model is trained by finetuning on both the retain and forget set, but reversing the gradient for the latter. There is a trade-off parameter $\beta$ that controls the relative importance of each of those two terms. As can be seen from Table 20, this induces a trade-off: smaller values of $\beta$ tend to forget more successfully (higher forget error), at the expense of poor performance (higher retain and test error), whereas lower values of $\beta$ lead to the opposite effect.

In our main experiments, we have carefully tuned this model in the same way that we do for SCRUBS: to achieve high error on the forget set while not sacrificing performance.

Next, to better understand trade-offs for both NegGrad and SCRUBS, we visualize the Pareto Front of each of these methods, in Figure 13. Our approach for this is the following. First, we vary hyperparameters for NegGrad (as seen in Table 20) and for SCRUBS (as seen in Table 22; see the next section on Sensitivity Analysis for SCRUBS for more details). Then, we construct the Pareto Front for each of these two methods as follows: from those tables, we include only the rows that represent (retain error, test error, forget error) points such that no other such point 'strictly dominates' it, i.e. no other point performs strictly better than it on all of those 3 metrics.

We draw the following conclusions from Figure 13:

1. SCRUBS' Pareto front contains many more points that are more effective at forgetting (higher forget error) compared to NegGrad's (visually: there are a lot of orange points to the right of blue points, in both subplots). Interestingly, **NegGrad can achieve high forgetting (as seen by the blue points in the top right corner), but this comes at the cost of retain error and test error for NegGrad (large y-values). On the other hand, SCRUBS' Pareto front contains points that have the best possible forget error, without sacrificing performance (bottom right corner of the plot).**

2. **Both NegGrad and SCRUBS have some points on their Pareto Front that our model selection procedure (that balances good forgetting with good performance on the retain set) would never choose as the model that we use for evaluation (e.g. to produce results for our tables).** In the case of SCRUBS, these points are the orange ones on the top right corner. These are included in the Pareto Front, because they have the best possible forget error, or are 'tied' with the point that does (so aren't strictly dominated by any other points), but they have poor retain error. Therefore, during model selection, we would instead choose the points on the bottom right corner, which have equally high forget er-

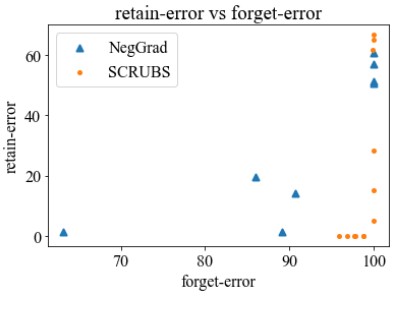

(a) retain vs forget errors.

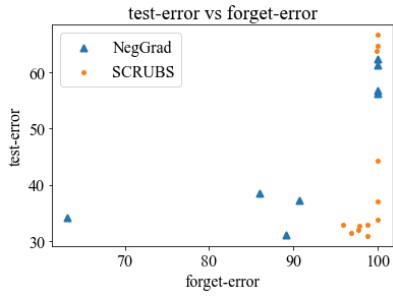

(b) test vs forget errors.

Figure 13: A visual comparison of the Pareto Front of NegGrad and SCRUBS. These points are obtained by Tables 20 and 22, which explore the performance of each of NegGrad and SCRUBS with respect to different hyperparameter settings. From those tables, we include only the rows that represent points of the Pareto Front. More concretely, a (retain error, test error, forget error) point is on the Pareto Front if no other such point 'strictly dominates' it, i.e. no other point performs strictly better than it on all of the 3 metrics.

ror but without sacrificing performance. In the case of NegGrad, some points of the Pareto Front fail to forget (bottom left corner, very low forget error), and similarly we would never choose those during model selection, since we can find other points in the Pareto front that have similar performance on the retain set but with higher forget error.

3. **Overall: SCRUBS' Pareto Front has significantly more points than NegGrad's Pareto Front on the most desirable 'corner' of the plots: the bottom right corner, which represents best forget error as well as best retain/test error.**

## A.8 SENSITIVITY ANALYSIS

Next, we analyze the sensitivity of SCRUBS to different hyperparameters that impact optimization.

As can be seen from Table 21, setting both $\alpha$ and $\gamma$ to 0 (which leads to only applying the max-step, to push the student away from the teacher on forget set examples) leads to high forget error, as expected, but significantly hurts performance. This is expected, since this variant has no incentive to retain useful information from the original model. Further, we observe that setting $\gamma$ to 3 (the best value that we found, which we used in our main results), can protect from this performance degradation, even if $\alpha$ is 0. On the other hand, setting only $\gamma$ to 0 leads to a worse performance than setting only $\alpha$ to 0, at least for the values considered in this table. Our general take-away, however, is that our results aren't overly sensitive to these hyperparameters. As long as they aren't set to 0, there is a wide range of them that yields similarly well-performing models.

## A.9 DETAILED COMPARISON AGAINST THE BAD-TEACHER APPROACH

In this section, we perform a more detailed empirical comparison against an idea presented in a recent concurrent work (preprint paper (Chundawat et al., 2022)). Their approach also utilizes a teacher-student framework, albeit in a different way. As in SCRUBS, the teacher is the original model trained on all available training data, and the student is initialized from the teacher and subsequently updated in a way that induces forgetting. However, the way in which this is achieved in their work is different than SCRUBS: in order to achieve forgetting, the student is encouraged to match the behaviour of a randomly-initialized model (a 'bad teacher') for the examples of the forget set (instead of pushing it away from the teacher for those examples as we do in SCRUBS).

Given that this idea is related to SCRUBS, we devote this section to a detailed discussion of the empirical comparison between these two. For this, we have re-implemented their approach (which we denote 'Bad-T', for 'bad teacher'), since the above preprint provides no available code for this model. We have tuned the learning rate and number of updates for 'Bad-T' thoroughly and perform model selection for this model in the same way that we do for SCRUBS.

| $\gamma$ | $\alpha$ | Test error | | Forget error | | Retain error | |
|---|---|---|---|---|---|---|---|
| | | mean | std | mean | std | mean | std |
| 0 | 0.1 | 44.78 | 4.76 | 94.78 | 1.55 | 23.11 | 9.45 |
| 1 | 0.1 | 31.33 | 2.68 | 96.89 | 0.57 | 0.00 | 0.00 |
| 2 | 0.1 | 31.89 | 2.86 | 97.67 | 0.27 | 0.00 | 0.00 |
| 3 | 0.1 | 30.89 | 2.04 | 98.78 | 0.63 | 0.00 | 0.00 |
| 4 | 0.1 | 30.78 | 2.18 | 99.33 | 0.54 | 0.00 | 0.00 |
| 5 | 0.1 | 31.22 | 2.04 | 99.33 | 0.54 | 0.00 | 0.00 |
| 6 | 0.1 | 30.89 | 2.28 | 99.67 | 0.47 | 0.00 | 0.00 |
| 7 | 0.1 | 31.22 | 2.53 | 99.67 | 0.47 | 0.00 | 0.00 |
| 3 | 0 | 31.33 | 3.30 | 100.00 | 0.00 | 0.00 | 0.00 |
| 3 | 0.1 | 31.00 | 2.18 | 98.78 | 0.63 | 0.00 | 0.00 |
| 3 | 1 | 30.22 | 1.23 | 91.44 | 2.69 | 0.00 | 0.00 |
| 3 | 2 | 30.44 | 2.22 | 84.67 | 3.27 | 0.00 | 0.00 |
| 3 | 3 | 31.33 | 1.78 | 81.44 | 4.24 | 0.00 | 0.00 |
| 3 | 4 | 30.67 | 2.05 | 79.11 | 4.13 | 0.00 | 0.00 |
| 3 | 5 | 31.00 | 1.78 | 76.89 | 4.61 | 0.00 | 0.00 |
| 3 | 6 | 31.67 | 2.23 | 75.22 | 4.99 | 0.00 | 0.00 |
| 3 | 7 | 31.56 | 1.64 | 74.22 | 5.19 | 0.00 | 0.00 |
| 0 | 0 | 64.78 | 2.67 | 100.00 | 0.00 | 65.11 | 2.20 |

Table 21: Sensitivity analysis for SCRUBS, for the experiment of large forget set, class unlearning on CIFAR-6 (forget 3 classes, i.e. 50%), with ResNet. Here we have analyzed the sensitivity of SCRUBS optimization to $\alpha$ (the hyperparameter that controls the influence of the term that minimizes the distance from the teacher on retain examples, in Eq. 6) and $\gamma$ (the hyperparameter that controls the influence of the cross-entropy term, in Eq. 6) on SCRUB's performance. The number of max-steps is set to 2 here, which is the best value discovered for this experiment.

| max-steps | $\gamma$ | $\alpha$ | Test error | | Forget error | | Retain error | |
|---|---|---|---|---|---|---|---|---|
| | | | mean | std | mean | std | mean | std |
| 1 | 0 | 0 | 56.67 | 3.21 | 30.56 | 7.05 | 43.78 | 6.92 |
| | 0 | 0.1 | 34.11 | 1.81 | 82.56 | 6.31 | 8.22 | 0.57 |
| | 1 | 0.1 | 32.89 | 2.99 | 95.89 | 2.57 | 0.00 | 0.00 |
| | 2 | 0.1 | 32.67 | 2.62 | 97.89 | 1.34 | 0.00 | 0.00 |
| | 3 | 0.1 | 32.78 | 2.51 | 98.78 | 0.57 | 0.00 | 0.00 |
| 2 | 0 | 0 | 64.78 | 2.67 | 100.00 | 0.00 | 65.11 | 2.20 |
| | 0 | 0.1 | 44.78 | 4.76 | 94.78 | 1.55 | 23.11 | 9.45 |
| | 1 | 0.1 | 31.33 | 2.68 | 96.89 | 0.57 | 0.00 | 0.00 |
| | 2 | 0.1 | 31.89 | 2.86 | 97.67 | 0.27 | 0.00 | 0.00 |
| | 3 | 0.1 | 30.89 | 2.04 | 98.78 | 0.63 | 0.00 | 0.00 |
| 3 | 0 | 0 | 66.67 | 0.00 | 100.00 | 0.00 | 66.67 | 0.00 |
| | 0 | 0.1 | 63.78 | 1.81 | 99.89 | 0.16 | 61.78 | 4.59 |
| | 1 | 0.1 | 44.33 | 4.91 | 100.00 | 0.00 | 28.22 | 8.87 |
| | 2 | 0.1 | 37.11 | 2.47 | 100.00 | 0.00 | 15.11 | 3.00 |
| | 3 | 0.1 | 33.78 | 2.11 | 100.00 | 0.00 | 4.89 | 1.97 |

Table 22: Additional sensitivity analysis for SCRUBS. This is an expansion of Table 21 that considers values for max-step aside from the best one.

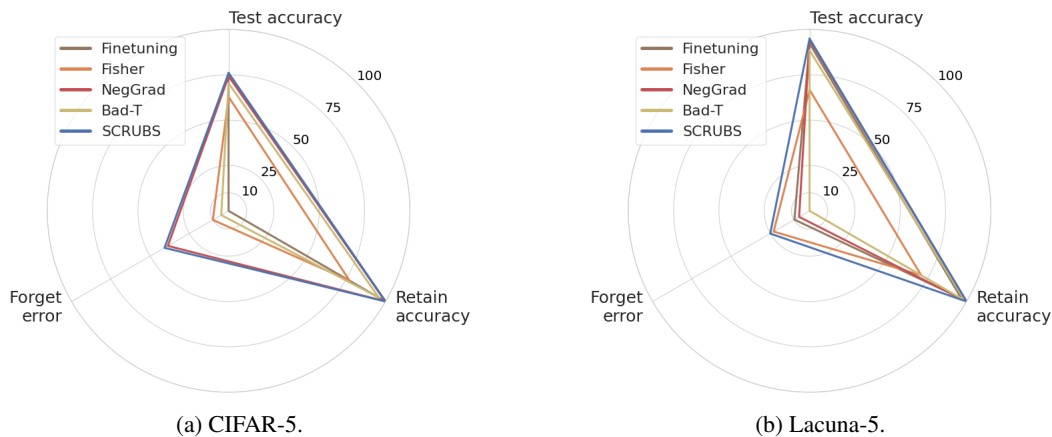

(a) CIFAR-5.      (b) Lacuna-5.

Figure 14: In the **small scale setting, selective unlearning**, SCRUBS is the only top-performer across settings. Bad-T performs poorly compared to other models. Each point represents the average across ResNet and All-CNN.

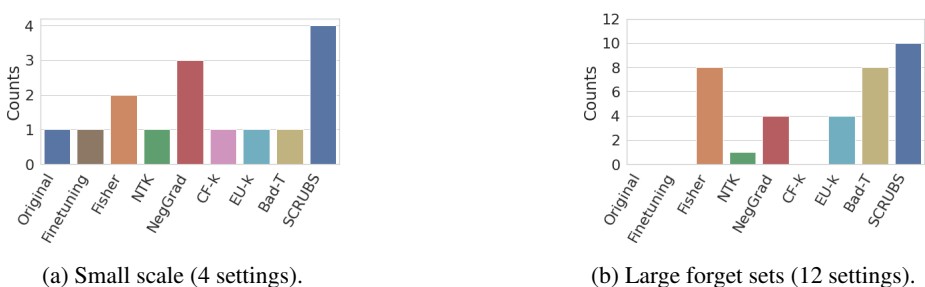

(a) Small scale (4 settings).      (b) Large forget sets (12 settings).

Figure 15: The number of times that each method was (one of the) top performer(s) in terms of the forget error, across all settings (architectures, datasets, selective vs class forgetting, forget set size, etc.). We consider a model a top-performer if its 95% confidence interval overlaps with that of the best mean. Small scale counts are over {ResNet, All-CNN} x {CIFAR, Lacuna} (4 total), and large forget set additionally x {class-50%, class-67%, selective-50%} (12 total). **SCRUBS is the most consistent in terms of its ability to unlearn**.

The results for Bad-T are shown in Tables 4, 5, 10, 11, 12, 13, 14 and 15 and and Figure 14, which is an adaptation of Figure 2 that includes Bad-T and excludes some other poorly-performing baselines to reduce clutter and aid in more clearly comparing Bad-T to other methods. We observe that the performance of Bad-T largely varies depending on the scenario, and that it fails to exhibit consistency in being a top-performer - in contrast to SCRUBS. In particular, we observe that it performs well in the scenarios with large forget sets (often in a tie with SCRUBS) but performs very poorly in the small scale regime (as can be seen clearly in Figure 14 for example).

To best understand its performance profile compared to SCRUBS, Figures 15 and 16 counts the times that a model is a top-performer. We consider a model to the a top-performer with respect to a metric if it has the best mean (over trials) performance over all other models considered, or if its confidence intervals overlap with the model's that has the best mean. These figures show (i) that SCRUBS is significantly more consistent in terms of successfully forgetting across a wide range of scenarios, and (ii) in addition, SCRUBS is more consistent in terms of not hurting the retain and test performance during unlearning.

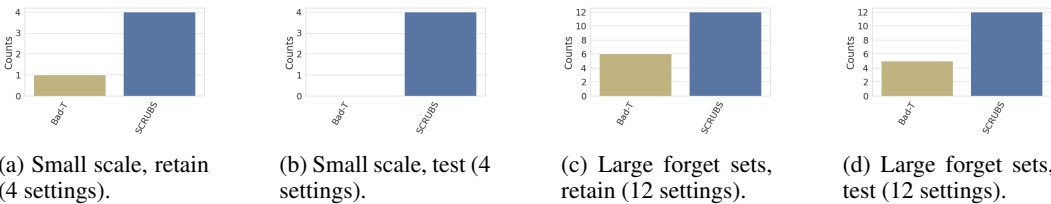

(a) Small scale, retain (4 settings).

(b) Small scale, test (4 settings).

(c) Large forget sets, retain (12 settings).

(d) Large forget sets, test (12 settings).

Figure 16: Focusing on the comparison between SCRUBS and Bad-T, we additionally show the number of times that each method was (one of the) top performer(s) in terms of the retain and test errors, across all settings (architectures, datasets, selective vs class forgetting, forget set size, etc.). We consider a model a top-performer if its 95% confidence interval overlaps with that of the model that has the best mean. Small scale counts are over {ResNet, All-CNN} x {CIFAR, Lacuna} (4 total), and large forget set additionally x {class-50%, class-67%, selective-50%} (12 total). **SCRUBS is more consistent compared to Bad-T in its ability to not hurt performance during unlearning**.

