# OpenReview forum: "The Brainy Student: Scalable Unlearning by Selectively Disobeying the Teacher"
_ICLR.cc/2023/Conference — Submitted to ICLR 2023_

### Official Review · Reviewer_eMqy · 2022-10-24

**Confidence:** 2
**Correctness:** 3
**Technical Novelty And Significance:** 3
**Empirical Novelty And Significance:** 3
**Recommendation:** 6

**Clarity, Quality, Novelty And Reproducibility:**

- Clarity: the paper is mostly clear. Figure 1 and 2 are a bit hard to read, as colors overlap.
- Novelty: the method seems intuitive, but I cannot judge the novelty as I'm not familiar with related work.
- Reproducibility: I see no code, but the appendix about implementation details seems comprehensive.

**Strength And Weaknesses:**

The method seems simple yet consistently effective, and there are many analyses and insights discussed in the experiment parts. It also points out problems of previous theoretically-inspired methods.

I'm not familiar with this domain, but seems the scales of experiments are modest, and even "large scale" setup is more about portion of forgetting instead of scale of model/data (as mentioned in intro)? The title, abstract, and intro might need to better characterize what is "scalable" in the method.

**Summary Of The Paper:**

The paper proposes a simple scheme for machine unlearning: learning to minimizing DL divergence with original model on the retain set (can also minimize test error), while maximizing DL divergence with original model on the forget set. It is shown to be effective across many task setups versus many baselines.

**Summary Of The Review:**

The method is simple and effective, but authors might want to better quantify the "scalable" part.

---

> ### Author Response · Authors · 2022-11-12
> **Response to Reviewer eMqy**
>
> We thank Reviewer eMqy for the positive feedback! We are glad to hear that the reviewer found our method “simple yet consistently effective” and that we provide many “analyses and insights” in our experiments.
>
> **RE: scalability** The reviewer makes a great point about qualifying scalability. Indeed, scalability was one of the major motivations behind our proposed method. In fact, other models don’t scale beyond simple datasets, and have instead proposed smaller architectures, to be able to run experiments (e.g. in the case of NTK). Other methods are able to handle larger datasets but run very slowly (in fact, in agreement with previous work, we found that Fisher is slower than the oracle of retraining from scratch) – this is shown in Figure 4.
>
> Our method is actually scalable in terms of all of these aspects: the scale of the model and data, as well as the proportion of the forget set. One way to see this is through Figure 4, in our largest-scale dataset considered, SCRUBS shows a large scale-up factor. This suggests that we can further increase the size of the dataset and model, which is important future work. However, in this paper we consider medium scale experiments indeed, as the reviewer pointed out, as otherwise we would be unable to compare against previous work. Further scaling up unlearning methods is crucial for practical applications and we believe that our work has made important strides in this direction. We included a discussion of this interesting point in Section A.1 in the Appendix.
>
> **RE: reproducibility** We also wholeheartedly agree with the reviewer about the need for reproducibility and code availability. Thus, we have made our code (anonymously) available here: https://anonymous.4open.science/r/SCRUBS-EE22
>
> We hope that our response addresses the reviewer’s concerns, and we would be very grateful if the reviewer can take another look.

---

### Official Review · Reviewer_8kmh · 2022-10-24

**Confidence:** 4
**Correctness:** 3
**Technical Novelty And Significance:** 2
**Empirical Novelty And Significance:** 2
**Recommendation:** 5

**Clarity, Quality, Novelty And Reproducibility:**

Clarity:
The writing and the presentation of the paper are very clear.

Quality:
The proposed method is sound and the given experiment results are solid. However, I have a few comments/questions:
1. As stated in the paper, the reported results are points on the Pareto front. I wonder how such a point is selected. Is it possible to compare various methods across the entire Pareto front to make the results more convincing? I.e., can we tune the parameters of the proposed methods as well as some baselines to get different trade-offs among the three evaluation metrics?
2. There are multiple ways to combine the three terms in the objective. E.g., we can use the ratio between the retaining loss and the forgetting loss, similar to a contrastive loss. I wonder if the authors have tried different forms of combinations and why the current linear form is chosen.
3. Sensitivity analysis over the hyperparameters and ablation studies could make the experimental results more convincing.

Novelty:
As discussed in the strengths v.s. weaknesses part, the novelty of the method is limited.

Reproducibility:
The method is simple to implement and many details of the experiments are given. I believe the reproducibility is decent.

**Strength And Weaknesses:**

Strengths:
1. The proposed method is relatively efficient and can apply to a larger set of unlearning samples compared to previous methods.

2. The proposed method is simple and easy to implement.

3. The experimental result shows that the proposed method has better consistency in giving the best performance among various tasks in terms of forgetting errors, retaining accuracy, and test accuracy.


Weaknesses:
1. The proposed objective is a minimax objective, and training for such an objective could be potentially very tricky and it may require a lot of tuning. The paper mentions that it follows certain techniques from training GAN models, but it still seems careful tuning is required to optimize the objective effectively. Also, it is hard to know whether the optimization result is good or bad without any theoretical understanding of the proposed objective.

2. The novelty of the proposed method is also limited. The proposed objective is similar to the consistency and contrastive loss used in self-supervised learning and the optimization procedure follows techniques of training GAN models.

3. The proposed method has better consistency over a variety of tasks. However, in some tasks, the proposed methods do not give the best performance and more investigation seems required to understand what kind of tasks the proposed method does better and why.

4. The proposed method does not have theoretical guarantees of any form. I.e., it is hard to know whether the unlearning set of samples has been truly forgotten. Showing theoretical results on deep models is very hard, and showing theoretical properties for simpler models could still be insightful.


**Summary Of The Paper:**

The paper studies the problem of machine unlearning. The paper proposes a method that trains a student model starting from the original model, and updates the student model based on the loss so that the student's loss on the unlearned samples is maximized while the loss on the retained samples is minimized. Different from previous machine unlearning work, the paper focuses on the efficiency and performance of deep neural networks, and shows improved results in terms of forgetting errors, retaining accuracy, and test accuracy.

**Summary Of The Review:**

Given the limited novelty and some questions about the current method and experiments, I think modifications and clarification need to be made to accept the current paper.

---

> ### Author Response · Authors · 2022-11-12
> **Response to Reviewer 8kmh (part 1)**
>
> We thank the reviewer for recognizing that our method is “simple and easy to implement”, has “better consistency in giving the best performance among various tasks”, and that “the writing and the presentation of the paper are very clear”.
>
> We address the reviewer’s comments below.
>
> **RE: min-max objective**. We agree with the reviewer that the difficulty in tuning this objective is one of the limitations of our method (we added a discussion of limitations in Section A.1 in the Appendix). We show an illustration of the training dynamics in Figure 5 (and Figures 9, 10 and 11 in the Appendix), to be as transparent as possible about the difficulties, and demonstrate how our proposed practical algorithm overcomes these difficulties in the scenarios we considered.
> In terms of theoretical understanding of the optimization objective, solutions have been proposed for simple convex-concave settings [1]. Even for nonconvex-concave settings, previous work provides convergence bounds [2,3]. A really interesting direction for future work is creating a simpler (e.g. convex-concave) variant of SCRUBS and studying it. However, this is a substantial step beyond our work and is outside of the scope of our paper. In the meantime, we want to emphasize that our practical algorithm for optimizing this objective does perform very well on a wide range of unlearning scenarios considered. While there is certainly room for improving both the understanding of this objective, as well as developing additional methods for its optimization, our work moves the field a significant step forward: SCRUBS is by far the most consistent method in delivering good unlearning performance across a wide range of scenarios.
>
> **RE: novelty is limited**. Indeed our method is inspired by contrastive losses, and teacher student methods, as we state in the paper. We do not consider it a weakness when well-known principles are adapted to make significant advances in a new area! We weren’t trying to reinvent the wheel in terms of the loss function used, nor the optimization algorithm used. But our contribution here is a different one: to adapt the framework/principles of teacher student and contrastive learning for a very different problem where it hasn’t been used before, leading to an approach for unlearning that is significantly different from any previously-published unlearning method. It is not trivial to adapt those frameworks for this problem (and indeed, it wasn’t done before), e.g. how should the teacher and student be defined in such a way that the goals of the unlearning problem are met? This is where the creativity and novelty of our work lies, and the resulting method is a significant advance for unlearning, as it leads to significant scalability and performance gains over previous methods.
>
> **RE: SCRUBS isn’t always a top-performer and we lack an understanding of when it’s best**. This is an excellent point! Our investigation has clearly shown that SCRUBS is by far the most consistent method in effectively forgetting, with a minimal decrease in performance, if any. However, SCRUBS is not the top-performer in every scenario and in terms of every metric (and this statement is true to a much larger extent for any other model). We believe that this nuanced picture actually is a contribution of this work and an indication of the extensiveness of our empirical evaluation (we went to great lengths to have a very comprehensive evaluation and inform the community of interesting trade-offs between methods), and also an indication of the complexity of the problem that we study.
>
> However, gaining a deeper understanding of all of the unlearning methods we consider (not just SCRUBS), their relative strengths and under which conditions we expect each to perform well, is crucial for continuing to make progress in this area. For instance, our investigation revealed a scenario (selective unlearning in cases with large forget set sizes) where NTK and Fisher (albeit performing very poorly in most cases) perform uncharacteristically well (see e.g. Figure 8 in the Appendix). This is a finding that we’ve uncovered that was not previously known and is surprising, in the sense that one couldn’t have trivially predicted this using known facts about these models. Given all this, we wholeheartedly agree with the reviewer that, as the field of machine unlearning matures, gaining some more understanding into the strengths and weaknesses of each method, and which factors impact their performance, is a very important research direction. This falls outside of the scope of our paper, but we’ve included a discussion of this in Section A.1 in the Appendix to encourage future work to investigate this direction.

---

> > ### Author Response · Authors · 2022-11-12
> > **Response to Reviewer 8kmh (part 2)**
> >
> > **RE: Lack of theoretical guarantees**. We agree with the reviewer that this is a limitation of our work, and we discuss this in Section A.1 in the Appendix. While theoretical guarantees are very important, they are hard to obtain in the case of deep networks (as is widely agreed and pointed out by the reviewer as well). Also, papers associated with guarantees, despite offering great insights, suffer from practical limitations. We aim to fill this gap in our work.
> > In our future work plans, one avenue is to strengthen the theoretical understanding of SCRUBS by providing bounds for the Mutual Information criteria we offer in Equation 2 and 3. In practice, exact mutual information is intractable. However, there is ongoing research in statistics and ML on the lower- and upper-bounds for mutual information between two spaces. Once such bounds are available, the optimization could be done by minimizing and maximizing those bounds. For instance, [4] introduces a variational lower bound between a teacher network and a student network and performs distillation by maximizing this bound. This method could be directly exploited to optimize Equation 3  in our work. But finding a tractable upper bound to minimize for Equation 2 is harder and an open problem (though it has recently attracted attention [5,6]). If more progress is made in this direction, we could measure the amount of the mutual information that remains between the two models. Nevertheless, we would like to re-emphasize that upper-bounding MI is hard and still an open problem that we leave for future work.
> >
> > However, in the absence of guarantees, we believe that a very extensive investigation of performance in terms of different metrics is crucial. Thus, the reviewer’s comment inspired us to further expand our investigation by considering a new set of metrics introduced in Goel et al. In these metrics, class confusion is introduced by mislabelling examples between two classes in the training set used to train the original model (e.g. class A examples are mislabelled as class B, and vice versa) and the role of unlearning is to resolve the class-confusion of the original model, in a setup where the forget set contains all and only the mislabelled examples, so successful unlearning results in resolving the class confusion. We devote section A.6 in the Appendix to describing these methods and evaluating SCRUBS and a number of baselines there. We find that SCRUBS performs very strongly even with this new set of metrics, by far better than any previous method, further enhancing its profile as a consistent top performer among all related works.
> > We believe that this consistency in delivering top performance across a large number of scenarios and metrics (this new metric is substantially different from the forget error, retain error and test error metrics we consider in the main paper) speaks to the usefulness of our work for the community. We thus argue that, despite the lack of theory, our work constitutes an important step forward.
> >
> > **RE: different points on the Pareto front**.  This is a very interesting comment and we agree it is important to investigate it. To that end, we have added a section in the Appendix (A.7) that discusses the trade-offs that may arise during the optimization process that may lead to different points in the Pareto front. We perform a case study there for the NegGrad model (see Equation 8), which is the second strongest model after SCRUBS in most settings considered. Its objective has a term that minimizes the loss on the retain set, and a term that maximizes the loss on the forget set. We show in Table 20 the trade-offs that arise based on how these two terms are weighted relative to each other during optimization. As expected, placing more weight on the former leads to better performance, at the expense of less effective unlearning, whereas placing more weight on the latter more effectively forgets, but degrades performance more. In our main experiments, we have carefully tuned this model in the same way that we do for SCRUBS: to achieve high error on the forget set while not sacrificing performance. But different practitioners that may prioritize one aspect over the other can set the balancing coefficient accordingly for their use case.
> >
> > **RE: different ways of combining terms in the loss**. Indeed, there is room for exploring different design choices there. In our work, we opted for simplicity (in terms of the form of the loss and and the number of hyperparameters introduced): since the linear formulation already delivered strong results, we looked no further. Future work can investigate other design choices which may lead to further improving SCRUBS’ performance.

---

> > > ### Author Response · Authors · 2022-11-12
> > > **Response to Reviewer 8kmh (part 3)**
> > >
> > > **RE: sensitivity analysis**. We thank the reviewer for this comment, and we have added more results to this effect in the Appendix. Overall, we have ablated different terms of our proposed objective and design choices of the optimization algorithm in Figure 5 (and Figures 9, 10 and 11 in the Appendix), we ablate the use of the cross-entropy term in Figure 12. We also performed a sensitivity analysis of SCRUBS’ hyperparameters, and added the results and a discussion in section A.8 (see Table 21 for results).
> > >
> > > **RE: reproducibility**. To further strengthen the reproducibility of our method, we have made our code anonymously available here: https://anonymous.4open.science/r/SCRUBS-EE22.
> > >
> > > **Overall**, we have worked very hard during the rebuttal based on the constructive feedback from the reviewer. In particular, we have explored trade-offs that can lead to different points in the Pareto front, performed a sensitivity analysis for SCRUBS’ hyperparameters, and, to strengthen our empirical evaluation given the lack of theoretical guarantees, we have experimented with a very different set of metrics, based on class confusion. We continue to observe throughout all experiments that SCRUBS is by far the most consistent method in delivering strong unlearning results with only minimal performance degradation. We thus believe that our work is an important step forward and a useful data point for the community. We would be very grateful if the reviewer takes another look and reconsiders their evaluation based on all the above.
> > >
> > > References:
> > > ==========
> > > [1] Tseng, P., 2008. On accelerated proximal gradient methods for convex-concave optimization. submitted to SIAM Journal on Optimization, 2(3).
> > >
> > > [2] Zhang, J., Xiao, P., Sun, R. and Luo, Z., 2020. A single-loop smoothed gradient descent-ascent algorithm for nonconvex-concave min-max problems. Advances in Neural Information Processing Systems, 33, pp.7377-7389.
> > >
> > > [3] Lin, T., Jin, C. and Jordan, M., 2020, November. On gradient descent ascent for nonconvex-concave minimax problems. In International Conference on Machine Learning (pp. 6083-6093). PMLR.
> > >
> > > [4] Ahn, S., Hu, S.X., Damianou, A., Lawrence, N.D. and Dai, Z., 2019. Variational information distillation for knowledge transfer. In Proceedings of the IEEE/CVF Conference on Computer Vision and Pattern Recognition (pp. 9163-9171).
> > >
> > > [5] Poole, B., Ozair, S., Van Den Oord, A., Alemi, A. and Tucker, G., 2019, May. On variational bounds of mutual information. In International Conference on Machine Learning (pp. 5171-5180). PMLR.
> > >
> > > [6] Cheng, P., Hao, W., Dai, S., Liu, J., Gan, Z. and Carin, L., 2020, November. Club: A contrastive log-ratio upper bound of mutual information. In International conference on machine learning (pp. 1779-1788). PMLR

---

> > > > ### Comment · Reviewer_8kmh · 2022-11-16
> > > > **Response to the rebuttal**
> > > >
> > > > Thanks authors for the detailed replies as well as the corresponding changes in the paper. The additional sensitivity analysis results and the more empirical evaluation to address some theoretical concerns definitely make the paper stronger. For the pareto front question though (again, thanks for the new table results), I wonder if it is possible to compare the pareto front of SCRUBS and NegGrad graphically? E.g., we may have two 2-D plots, one showing the trade-offs between test error and forget error, and the other one for retain error and forget error.

---

> > > > > ### Author Response · Authors · 2022-11-18
> > > > > **Thank you for the positive feedback, we have created the additional plots you requested**
> > > > >
> > > > > Dear reviewer 8kmh,
> > > > >
> > > > > Thank you so much for responding to us and for the additional feedback. We are glad that you agree that the improvements we have made during the rebuttal make the paper stronger!
> > > > >
> > > > > That’s a great suggestion. We have created these additional plots and added them to the Appendix section A.7, in Figure 13. We have elaborated in that section on the experimental setup for this and the take-aways from this plot.
> > > > >
> > > > > We summarize here our findings:
> > > > > -  SCRUBS' Pareto front contains many more points that are more effective at forgetting (higher forget error) compared to NegGrad's. Interestingly, **NegGrad can achieve high forgetting , but this comes at the cost of retain error and test error for NegGrad. On the other hand, SCRUBS' Pareto front contains some points that have the best possible forget error, without sacrificing performance**.
> > > > >
> > > > >
> > > > > - **Both NegGrad and SCRUBS have some points on their Pareto Front that our model selection procedure (that balances good forgetting with good performance on the retain set) would never choose**. In the case of SCRUBS, these points are the orange ones on the top right corner of each subplot of Figure 13. These are included in the Pareto Front, because they have the best possible forget error, or are `tied' with the point that does (so aren't strictly dominated by any other points), but they have poor retain error. Therefore, during model selection, we would instead choose the points on the bottom right corner, which have equally high forget error but without sacrificing performance. In the case of NegGrad, some points of the Pareto Front fail to forget (bottom left corner), and similarly we would never choose those during model selection, since we can find other points in the Pareto front that have equally good retain error but with higher forget error.
> > > > >
> > > > >
> > > > > - **Overall: SCRUBS' Pareto Front has significantly more points than NegGrad's Pareto Front on the most desirable 'corner' of the plots: the bottom right corner, which represents best forget error as well as best retain/test error.**
> > > > >
> > > > > We hope that this addresses your suggestion and that you will consider raising your score in light of all of the changes we have made during the rebuttal.

---

### Official Review · Reviewer_GpWt · 2022-10-26

**Confidence:** 3
**Correctness:** 2
**Technical Novelty And Significance:** 2
**Empirical Novelty And Significance:** 2
**Recommendation:** 5

**Clarity, Quality, Novelty And Reproducibility:**

* The paper is not well written
* The paper is rather hard to read (it contains a lot of grammatical errors) and difficult to connect sentences.
* This proposed approach of using student-teacher architecture for machine unlearning is not entirely new.
* Typos:
Abstract: motivated to (motivated by)
Page 8: state-the-art (state-of-the-art)



**Strength And Weaknesses:**

Strengths:
* The idea of machine unlearning for larger scales, by simultaneously encouraging the student to ‘stay close to the teacher on retaining examples, while encouraging it to ‘move away from the teacher on forgetting examples, is an interesting contribution to the different domains of AI.
* The problem formulation and its components are well explained with mathematical equations.

Weaknesses:
* This proposed approach of using student-teacher architecture for machine unlearning is not entirely new. There is a plethora of papers in machine unlearning that uses a student-teacher approach to train the models.
* The proposed framework is similar to the recent work “Can Bad Teaching Induce Forgetting? Unlearning in Deep Networks using an Incompetent Teacher“, Chundawat et al. 2022. However, the authors have not cited this work, and experimental analysis is limited to previous works.
* Figure 3 requires standard error bars for better comparison across all the methods. Also, the authors could perform a statistical significance test to compare the results.


**Summary Of The Paper:**

The paper proposes a machine unlearning approach, Brainy student, to forget a subset of training data even at larger scales. The solution is inspired by a recent popular student-teacher architecture where a student network disobeys the teacher network to avoid inheriting information about the forget set while obeying the teacher network for the retain set. The experimental investigation reveals that the proposed method significantly improves upon previous methods in achieving unlearning in a wide range of scenarios.

**Summary Of The Review:**

Overall, the authors propose a new framework, brainy student, a student-teacher network-based model for the purpose of machine unlearning at larger scales. However, the paper lacks model training details, lacks a discussion on why the proposed model is superior to the previously existing methods, and the robustness of the model is still to be validated.

---

> ### Author Response · Authors · 2022-11-12
> **Response to Reviewer GpWt**
>
> We thank reviewer GpWt for recognizing that our proposed method “significantly improves upon previous methods in achieving unlearning in a wide range of scenarios”, that our work “is an interesting contribution to the different domains of AI” and that our formulation is “ well explained with mathematical equations”. We reply to the reviewer’s comments below.
>
> **RE: “there is a plethora of papers in machine unlearning that uses a teacher-student approach”.** We are not aware of any other ones, aside from (Chundawat et al) that the reviewer pointed us to (see next point below). We would be very grateful if reviewer GpWt could point us to that plethora of papers that we missed.
>
> **RE: Chundawat et al.** We thank reviewer GpWt for pointing us to that related paper. However, we really hope that the reviewer considers that: 1) that paper is an arXiv preprint and has not been published! Therefore, we don’t think it’s fair to say that concurrent unpublished work deducts novelty points from our paper. 2) the paper does not make code available. 3) that paper itself makes no comparisons against state-of-the-art unlearning methods unfortunately (mostly compares against versions and ablations of itself) and 4) the way in which the problem is cast into teacher-student framework in our case is substantially different than in theirs (e.g they use two teachers, a bad and a good one, whereas we only use one). This difference in casting into a teacher-student framework makes a big difference in performance - please see below.
>
> Nevertheless, we agree to include this paper in our related work section. We have also gone to great lengths to reimplement this method ourselves and tune it as well as possible, based on the details in their paper. We devote a section in the Appendix (Section A.9) to a detailed discussion between SCRUBS and this method, which we refer to as Bad-T (for ‘bad teacher’), and we ran extensive experiments to compare against it. Please see Section A.9 in the Appendix for all details.
>
> We emphasize the take-away from that section here: while Bad-T is competitive in some isolated scenarios (as are some of the other baselines/competing approaches evaluated in our work), there are cases in our extensive evaluation framework where it performs very poorly. Overall, it significantly lags behind SCRUBS in terms of its consistency to unlearn across a wide range of different scenarios, without hurting performance, as shown in Figures 13, 14 and 15.
>
> **RE: statistical test.** To construct Figure 3, we indeed ensure that the best performer is one with a statistically significant difference. Specifically, each model on each scenario is run for 3 different trials (random seeds, as done in competing papers), and the standard deviation is computed. From this, we derive the standard error and associated z-values, leading to 95% confidence intervals. We then decide when a model is a better performer than another model by inspecting not only the mean but also their confidence intervals. If the confidence intervals overlap, we consider both methods as equally good. In Figure 3 we merely show the *counts* of how many times a model is a top performer (where top performer is defined as having the best mean, or having a worse mean but with overlapping confidence intervals against the best performer.
>
> **RE: “the paper is not well-written and hard to read”.** We were surprised at this comment, as it contradicts what all of the remaining reviewers thought, namely that our paper is “easy to follow”, “very clear” (Reviewer VJBS),  “The writing and the presentation of the paper are very clear” (Reviewer 8kmh) and “the paper is mostly clear” (Reviewer eMqy). We would be grateful if reviewer GpWt took another look, or gave us constructive feedback for how to further improve the quality of our writing.
>
> Overall, we have addressed the comments of the reviewer, and strengthened our paper as a result. We would be very grateful if the reviewer can reconsider their judgment of our work in light of our rebuttal (e.g. our extensive comparisons against Chundawat et al and clarifications).

---

### Official Review · Reviewer_VJBS · 2022-10-27

**Confidence:** 3
**Correctness:** 3
**Technical Novelty And Significance:** 3
**Empirical Novelty And Significance:** 3
**Recommendation:** 6

**Clarity, Quality, Novelty And Reproducibility:**

This paper is very clear and properly discusses previous literature. The authors correctly model the problem of interest and present a new method for it.

Unfortunately, the authors do not include material to reproduce the results.


**Strength And Weaknesses:**

Strengths

- This paper is easy to follow and well-motivated.
- In the context of new legislation and other relevant use cases, this paper has a clear motivation.
- The proposed method seems to be practical.

Weaknesses

- It is a relevant point to include the source code and enable others to reproduce the experiments.
- The current method is valid to other architectures, such as Vision Transformers? Or a new deep unlearning method needs to be envisioned? This paper deserves this discussion.
- Given the current metrics used, I did not have a clear understanding of issues related to, for instance, class imbalance.
- The evaluation metrics should be clearly presented, such as using equations.
- Besides the points highlighted as future work, the authors should discuss the limitations of their method. Given the space restriction, some parts of related work could go to the Appendix.

**Summary Of The Paper:**

The focus of this paper is deep machine unlearning. This problem refers to removing the influence of a subset of data from the weights of a trained deep model. For instance, an application of this kind of method is to remove user data to guarantee their "right to be forgotten." Other applications include removing noisy labels, outliers, harmful biases, and out-of-date instances. Given the widespread use of DNNs, this is a significant problem.

Since previous approaches to deep machine unlearning make unrealistic assumptions related to DNNs, such as assuming the convexity of loss functions, there is a need for scalable solutions. The student-teacher methodology inspires the proposed solution. Here, the student ignores to-forget instances and considers all others.

A hypothesis of this paper is that by leveraging the student-teacher-inspired methodology are able to scale and simultaneously provide minimal performance loss. The authors designed a new method based on that methodology and conducted empirical experiments to validate their hypothesis. All research questions are clearly stated.

Specific comments:

- I have difficulty understanding why, in practice, Eq. 4 does not work. I suggest the authors include some intuition on that.
- In Sec. 3, Parag. 5, "performing this maximization step" ->  "performing this minimization step" (according to Eq. 4).

**Summary Of The Review:**

This paper presents a scalable solution to the deep unlearning method by exploiting the student-teacher methodology. The results validate their hypothesis. Additionally, there are missing important discussions and missing original code implementation.

---

> ### Author Response · Authors · 2022-11-12
> **Response to Reviewer VJBS (part 1)**
>
> We thank the reviewer VJBS for recognizing that our paper is “easy to follow”, “very clear” and our method is “well-motivated” and a “scalable solution”.
>
> As per the reviewer’s suggestions, we have made the following modifications: 1) we uploaded our code anonymously at https://anonymous.4open.science/r/SCRUBS-EE22, 2) we have added the detailed equations describing the metrics used, in the Appendix (Equation 7 for instance), and 3) We have included a subsection of the Appendix (Section A.1) with extensive discussion of the limitations of our work and interesting directions for future work (see more details below).
>
> We reply to the reviewer’s remaining comments and questions below.
>
> **RE: why Equation 4 does not work in practice.** Equation 4 only maximizes the distance (minimizes the negative distance) between the student and the teacher on the forget set examples. This indeed leads to effective forgetting, but significantly degrades the performance on the retain and held-out sets of examples (as we show in Figure 5a). The intuition is the following: pushing the student away from the teacher on the forget set examples can lead to the student moving in a number of different directions, some of which may lead to poor results on the retain and held-out sets, and Equation 4 is agnostic to that. Our intuition for further including a term in Equation 5 that pulls the student close to the teacher for the retain examples, is to constrain the directions that the student can move in during optimization, in order to only consider those that both satisfy achieving forgetting as well as maintaining useful knowledge. This protects the performance on the retain set, and the generalization ability of the model. We are happy to discuss further if this is still not clear.
>
> **RE: “performing this maximization step”**: the reviewer is correct that it can also be seen as a minimization step. We use the term maximization to refer to the distance between the student and teacher. This is maximized, as the objective is to *minimize* the *negative* distance. Thank you for pointing this out. We have rephrased that sentence in the paper to clarify.
>
> **RE: using our method with other architectures such as Vision Transformers**: This is an excellent question! In principle, our method (and previous unlearning algorithms too) are architecture-agnostic and can be adapted to work with any backbone. The choice of architecture might be application dependent in practice – e.g. vision transformers aren’t state-of-the-art yet for image classification tasks, but in NLP, Transformer architectures have been a crucial ingredient in recent progress. Given this, it would be very interesting to investigate how different unlearning algorithms interact with the inductive biases present in the architectures themselves. Since machine unlearning is still a very young area of research, this hasn’t been explored yet in any paper (in fact, previous work tends to make architectures smaller due to scalability difficulties, which our method overcomes). Here, we utilize the same architectures (i.e. ResNet and All-CNN) as in previous work to facilitate comparisons and we think it’s fair to say that investigating additional significantly different architectures is beyond the scope of this particular paper, but it’s a really interesting thread of future work, which we included in Section A.1.
>
> **RE: class imbalance**, this is a great point too. To date, no previous method has systematically studied the effect of class imbalance on unlearning methods. Following previous work, the datasets we use are class-balanced. But introducing imbalance opens several questions. For example, is it easier to unlearn a class if it’s a class that has only a few examples in the training set, compared to a class that has a large number of training examples? Our selective unlearning experiments touch on this to some extent, in that we observe selective forgetting (forgetting just a subset of a class) to be much harder than class forgetting (forgetting all examples of a class), across the board for more methods. Further, imbalance in the dataset may pose challenges for evaluation purposes too and we may need to rethink the metrics accordingly to make them more robust. This is a really interesting direction for future work, we have included it in Section A.1.

---

> > ### Author Response · Authors · 2022-11-12
> > **Response to Reviewer VJBS (part 2)**
> >
> > **Re: limitations**. A more extensive discussion is a great suggestion. We have added a section in the Appendix (Section A.1) discussing limitations and additional future work directions, including your insightful comments throughout the review. To summarize here, there are two most prominent limitations of SCRUBS: 1) the lack of theoretical guarantees. While theoretical guarantees are great, they are hard to obtain in the case of deep networks (as is widely agreed and pointed out by another reviewer as well). Also, papers associated with guarantees, despite offering great insights, suffer from practical limitations. We aim to fill this gap in our work, but look forward to insights in future work for striking a good compromise between good performance, effective unlearning, scalability and theoretical insights. 2) The difficulty in tuning the min-max objective of SCRUBS (as is widely recognized for similar losses, as in GANs). For instance, this objective can lead to oscillating behaviour (as we show in Figure 5 for example). We remedy this by providing an algorithm that works very well in practice, and yields results that are by far the most consistent among previous works in achieving forgetting with only a minimal decrease in performance (if any). However, there is certainly room in improving the stability and effectiveness of this optimization in the future.
> >
> > **Overall**: we hope that the reviewer recognizes that we’ve resolved the reproducibility concerns (we made our code available) and discussed the limitations and interesting future work more extensively, and clarified all points mentioned in the review. Given all this, we would be very grateful if the reviewer can have another look.

---

### Author Response · Authors · 2022-11-16
**Dear reviewers, are there any outstanding concerns about our paper?**

Dear reviewers,

Since the deadline is approaching after which we can no longer interact with you, we would be very appreciative if you could take a look at our responses and let us know if there are any other outstanding concerns, so that we have the chance to respond to those.

We have worked hard so far to address your feedback, which resulted in changes to the paper (we added a new set of metrics for evaluation, sensitivity analysis and analysis of trade-offs that can occur during optimization, we compared against an additional baseline and added a new section to discuss additional limitations and future work). We also released our code, as per the request of several reviewers. Our paper has significantly improved as a consequence. We believe that through these changes we have addressed the concerns raised in the initial reviews, and we would love to hear back from the reviewers and have the chance to respond to any remaining concerns.

We thank you again for all the time and energy you have already spent in reviewing this paper.

---

### Decision · Program_Chairs · 2023-01-20

**Decision:**

Reject

**Justification For Why Not Higher Score:**

See the aforementioned main concern (A).

**Justification For Why Not Lower Score:**

N/A.

**Metareview: Summary, Strengths And Weaknesses:**

The main contribution of this work lies in proposing a simple, scalable approach to deep machine unlearning by exploiting the student-teacher framework with a flavor of contrastive learning.

After reviewing the authors' rebuttal and an active discussion, the reviewers have agreed on the (A) main concern of the lack of theoretical analysis in terms of the unlearning behavior and the convergence of the unlearning process. For example, it is not clear why encouraging the student to 'move away' from the teacher on forget examples is an appropriate (sub)criterion (i.e., last/third term in equation 6) since there is no assumption in the problem that forget examples or the input space they reside in cannot be predicted well. Optimizing this third term seems to imply poor prediction on them. One may argue that the authors can meticulously tune the two hyperparameters in equation 6 to mitigate this. However, doesn't this come at a significant computational cost? Having a rigorous theoretical analysis would address this concern.

As a minor note, if we look at their final training objective in equation 6, the middle/second term/subcriterion captures exactly the loss function over the retain set or the loss function needed to retrain from scratch. In this regard, the time incurred per update would be the same. The only difference is probably that the authors start with weight initializer w^o for their proposed algorithm (Algorithm 1) and when they retrain from scratch, they do a random initialization. In the latter case, would there still be a scalability benefit if they start with weight initializer w^o? The authors can consider clarifying this in their revised paper as well.

**Summary Of Ac-Reviewer Meeting:**

As mentioned in the meta-review, the reviewers in attendance have agreed on the main concern (A) stated above, which has led to the final recommendation/decision.